# Resolving the cause of recurrent *Plasmodium vivax* malaria probabilistically

Aimee R. Taylor [1,2,8]*, James A. Watson [3,4,8]*, Cindy S. Chu[4,5], Kanokpich Puaprasert[6], Jureeporn Duanguppama[3], Nicholas P.J. Day[3,4], Francois Nosten [4,5], Daniel E. Neafsey[2,7], Caroline O. Buckee[1], Mallika Imwong [3,6] & Nicholas J. White[3,4]*

Relapses arising from dormant liver-stage *Plasmodium vivax* parasites (hypnozoites) are a major cause of vivax malaria. However, in endemic areas, a recurrent blood-stage infection following treatment can be hypnozoite-derived (relapse), a blood-stage treatment failure (recrudescence), or a newly acquired infection (reinfection). Each of these requires a different prevention strategy, but it was not previously possible to distinguish between them reliably. We show that individual vivax malaria recurrences can be characterised probabilistically by combined modelling of time-to-event and genetic data within a framework incorporating identity-by-descent. Analysis of pooled patient data on 1441 recurrent *P. vivax* infections in 1299 patients on the Thailand–Myanmar border observed over 1000 patient follow-up years shows that, without primaquine radical curative treatment, 3 in 4 patients relapse. In contrast, after supervised high-dose primaquine only 1 in 40 relapse. In this region of frequent relapsing *P. vivax*, failure rates after supervised high-dose primaquine are significantly lower (~3%) than estimated previously.

[1] Center for Communicable Disease Dynamics, Department of Epidemiology, Harvard T. H. Chan School of Public Health, Boston, MA 02115, USA. [2] Broad Institute of MIT and Harvard, Cambridge, MA 02142, USA. [3] Mahidol-Oxford Tropical Medicine Research Unit, Faculty of Tropical Medicine, Mahidol University, Bangkok 10400, Thailand. [4] Centre for Tropical Medicine and Global Health, Nuffield Department of Medicine, University of Oxford, Oxford OX3 7FZ, UK. [5] Shoklo Malaria Research Unit, Mae Sot, Tak Province 63110, Thailand. [6] Department of Molecular Tropical Medicine and Genetics, Faculty of Tropical Medicine, Mahidol University, Bangkok 10400, Thailand. [7] Department of Immunology and Infectious Diseases, Harvard T. H. Chan School of Public Health, Boston, MA 02115, USA. [8] These authors contributed equally: Aimee R. Taylor, James A. Watson. *email: ataylor@hsph.harvard.edu; jwatowatson@gmail.com; nickwdt@tropmedres.ac

*P*lasmodium vivax is the most geographically widespread cause of human malaria, with an estimated 2.5 billion people at risk of infection[1]. Vivax malaria is characterised by its ability to relapse following activation of dormant liver-stage parasites called hypnozoites. Multiple relapses can follow a single mosquito inoculation[2]. However, recurrent vivax malaria can also be caused by recrudescence (resulting from blood-stage treatment failure) or, within the endemic area, by reinfection (from a new mosquito inoculation). Distinguishing between these different causes of recurrent blood-stage infection is challenging, and has historically relied on the inherent periodicity of vivax relapse (time-to-event). Two distinct relapse phenotypes have been described, both exhibiting strong periodicity. In temperate climes the interval from primary infection to relapse is often long (circa 9 months), while in tropical climes relapses occur frequently at short intervals (3–4 weeks after treatment with rapidly eliminated antimalarial drugs)[3,4]. Therefore, time-to-recurrence—the interval since treatment of the preceding episode—provides valuable information. The probability and timing of recrudescent *P. vivax* infections depends upon parasite drug susceptibility, parasite biomass at the start of treatment, drug pharmacokinetics, and host immunity. Simple intrahost pharmacodynamic models of malaria argue that relapse will preempt recrudescence of tropical *P. vivax* when resistance is low grade[3]. Reinfection rates are usually seasonal.

Standard antimalarials recommended for the treatment of vivax malaria (e.g. chloroquine) act on blood-stage parasites only and are not hypnozoiticidal. The only generally available drug that kills hypnozoites (radical cure) is the 8-aminoquinoline primaquine. Although recommended in most endemic countries, primaquine is not widely used outside of South America because of the risks of iatrogenic haemolysis in patients with glucose-6-phosphate dehydrogenase (G6PD) deficiency[5]. Tafenoquine, also an 8-aminoquinoline, is approved in some countries but is yet to be deployed. The inability to distinguish between relapse, recrudescence, and reinfection complicates our understanding of *P. vivax* biology, epidemiology, and the assessment of therapeutic interventions. In endemic areas, estimates of the failure rates of radical curative drugs will be biased upwards by reinfections. The inability to distinguish between relapse and recrudescence contaminates failure estimates of blood-stage treatments[6]. Genetic analyses complement time-to-recurrence data by characterising the genotypes of parasites across successive blood-stage infections. However, in contrast to *P. falciparum*, where genotyping helps identify recrudescence versus reinfection with a new unrelated parasite[7–9], genotype information is harder to interpret in *P. vivax*. First, because *Plasmodium* parasites are capable of both self- and cross-fertilisation[10], the types of genetic relationships compatible with relapsing parasites are complex, including clones, siblings, and unrelated parasites. Second, over repeated *P. vivax* inoculations, individuals living in an endemic area can amass a bank of genetically diverse hypnozoites[11,12], which can relapse independently. Genetically unrelated *P. vivax* parasites across primary and recurrent infections are thus compatible with both reinfection and relapse, e.g. parasites isolated from a relapse can be unrelated either because a recent inoculation contained unrelated parasites, or because they derive from the genetically diverse bank of liver hypnozoites (i.e. an infection that predates the most recent inoculation)[3,6,11–17]. Thus, until now it has not been possible to characterise accurately therapeutic responses in vivax malaria in endemic areas.

In this study, we estimate individual-level recurrence states using statistical models of time-to-event and genetic data from two large randomised controlled trials conducted on the Thailand–Myanmar border. The time-to-event model accounts for the various treatments administered, thereby capturing their different pharmacokinetic and pharmacodynamic properties. For a given individual, the genetic model considers the relationships of parasites within and across all infections. Each relationship has an expected relatedness. We define relatedness as the pairwise probability of identity-by-descent (IBD), thus accounting for unrelated parasites that are identical-by-state (IBS) due to chance sharing of common alleles. Using inferred individual-level recurrence states, we characterise relapse dynamics and compute the reinfection-adjusted failure rate of high-dose (total dose of 7 mg/kg) primaquine on the Thailand–Myanmar border. We give recommendations for genetic marker requirements in future studies in similar settings.

## Results

**Overview of analysis**. Individual-level data from two large clinical studies in acute vivax malaria patients presenting to clinics on the Thailand–Myanmar border were pooled (VivaX History trial: VHX[18], Best Primaquine Dose trial: BPD[19]). An overview of each trial is given in Table 1. Unless stated, results are based on data from both trials combined. These pooled data totalled 1299 patients (primary episodes) with 1441 recurrences observed over 1007 patient follow-up years. Time-to-event data (2708 observations including right-censored intervals) were analysed using a Bayesian generative population mixture model accounting for

**Table 1 Summary of the VHX (columns one to three) and the BPD (columns four and five) trial features.**

|  | AS | CQ | PMQ + CQ | PMQ + CQ | PMQ + DP |
|---|---|---|---|---|---|
| Number of patients, *n* | 224 | 222 | 198 | 329 | 326 |
| Male patients (%) | 71 | 65 | 64 | 66 | 63 |
| Median patient age [years] (range) | 19 (1.5–62) | 18 (1.5–62) | 18 (2–63) | 20 (1.5–61) | 20 (1.3–63) |
| Median days follow-up (range) | 314 (1–393) | 364 (11–412) | 364 (0–383) | 364 (7–415) | 364 (7–685) |
| Geometric mean parasites per µL at enrolment (range) | 4113 (144–47,665) | 3920 (168–44,211) | 3524 (136–62,298) | 4134 (32–59,032) | 3726 (48–381,824) |
| Number of patients whose infections recurred, $n_r$ (% of *n*) | 177 (39) | 165 (36) | 35 (8) | 47 (10) | 34 (7) |
| $n_r$ with one or more paired recurrence genotyped and analysed (% of $n_r$) | 11 (6) | 88 (53) | 32 (91) | 46 (98) | 31 (91) |
| Number of recurrences, *r* | 722 | 587 | 40 | 53 | 39 |
| Number of paired recurrences genotyped and analysed (% of *r*) | 11 (2) | 355 (60) | 34 (85) | 52 (98) | 35 (90) |

Treatment allocation was randomised in both trials. Genetic data are reported in terms of the number of paired recurrences genotyped and analysed, since episode pairs are required for recurrence state inference and some paired recurrences were not analysed due to computational complexity. Five patients had unpaired thus unanalyzable genetic data. Data from six paired recurrences were omitted due to computational complexity

*AS* artesunate, *CQ* chloroquine, *PMQ* primaquine, *DP* dihydroartemisinin-piperaquine

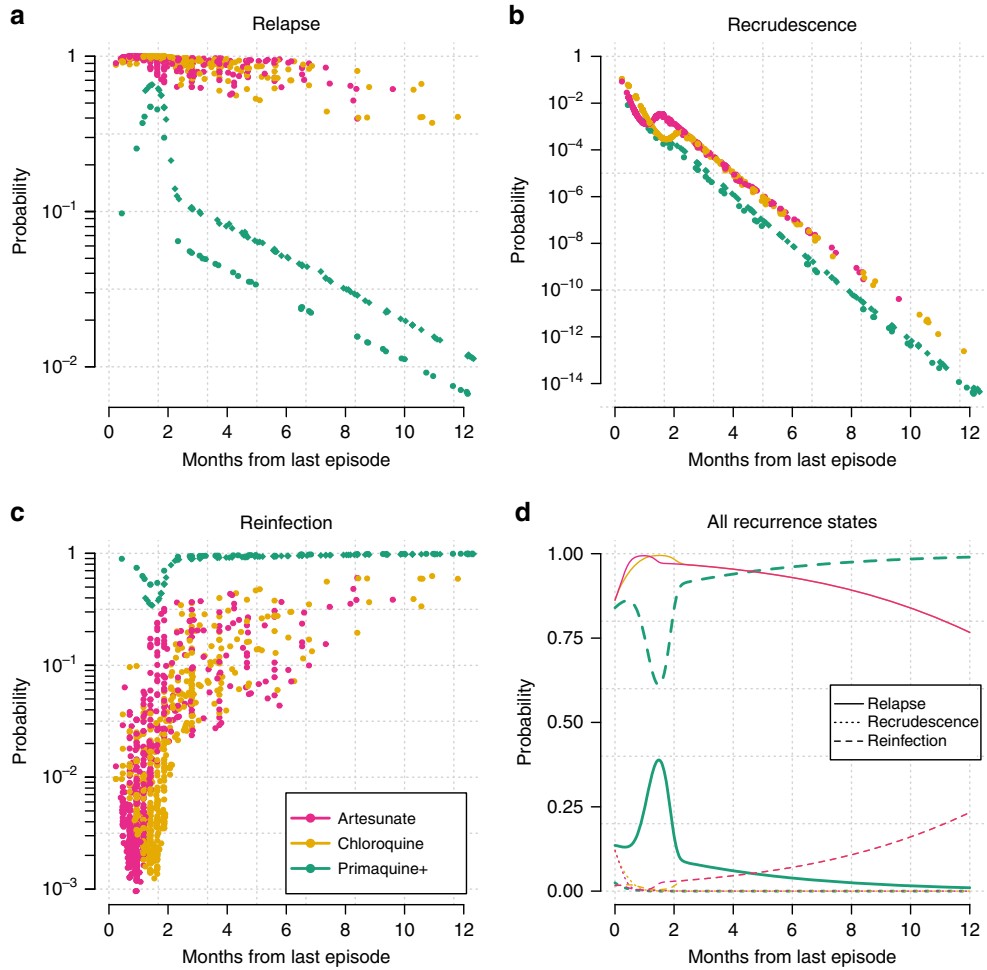

**Fig. 1 Dynamics of *P. vivax* relapse, recrudescence, and reinfection.** Estimates generated under the time-to-event model of the probabilities ($\log_{10}$ scale) of relapse (**a**), recrudescence (**b**), and reinfection (**c**) for all observed recurrences ($n = 1441$) are shown as a function of the interval since the last episode of vivax malaria (VHX trial: dots; BPD trial: diamonds). Colours correspond to the treatment used in the previous episode, where Primaquine+ refers to high-dose primaquine with a partner drug (either chloroquine or dihydroartemisinin-piperaquine). For each treatment (colour), the population mean probabilities (linear scale) for the three recurrence states (solid, dotted and dashed lines) are shown as a function of time-to-recurrence (**d**).

antimalarial drug treatment. Time-to-relapse was modelled as a mixture of periodic events (Weibull distributed) and constant-rate events (exponentially distributed). Times to recrudescence and reinfections were also modelled with constant rates using strongly informative priors for identifiability. The time-to-event model outputs individual probabilities for each of the three recurrence states, used subsequently as prior estimates for the genetic model. The genetic model incorporated *P. vivax* microsatellite marker data using a novel, taylor-made, probabilistic framework of genetic relationships. Its output was the final probabilities of relapse, recrudescence, and reinfection, used to re-estimate the failure rate of high-dose primaquine on the Thailand–Myanmar border, where a failure is defined as either relapse or recrudescence, but not a reinfection. Further analysis of simulated data using the genetic model gave generalisable estimates for the number of polymorphic microsatellite markers needed to determine accurately the unknown recurrence states based on genetic data alone.

**Dynamics of recurrence states**. Because of the periodicity of relapse, and the rarity of recrudescence in this setting, the interval between successive episodes of *P. vivax* is highly informative. Figure 1 shows the mean posterior probabilities of the recurrence states generated under the time-to-event model. The probability of reinfection varies over three orders of magnitude as a function of the time since last episode conditional on the treatment received. Following radical curative treatment with high-dose primaquine and partner drug, a recurrence in the first few months has a higher probability of being a relapse, and subsequent recurring infections have a higher probability of being reinfections. The dynamics of time-to-relapse, irrespective of whether primaquine was administered, were explained as a 60/40 mixture between periodic and constant-rate events (95% credible interval (CI) for the periodic component: 57–65, corresponding to the parameter $q$ which is given on the logit scale in Supplementary Table 1). The constant-rate component is consistent with occasional relapses occurring randomly (i.e. not periodically) many months after the primary infection with a short-latency phenotype *P. vivax*[2].

The posterior uncertainty intervals for the individual probabilities of relapse for each recurrence are shown in Fig. 2. For approximately 75% of the recurrences observed after treatment without high-dose primaquine, the posterior distributions were extremely narrow with the probabilities of relapse very close to 1 (Fig. 2a). The remaining 25% had relapse posterior probabilities greater than 0.3 but with wide credible intervals. For the recurrences observed after high-dose primaquine, approximately

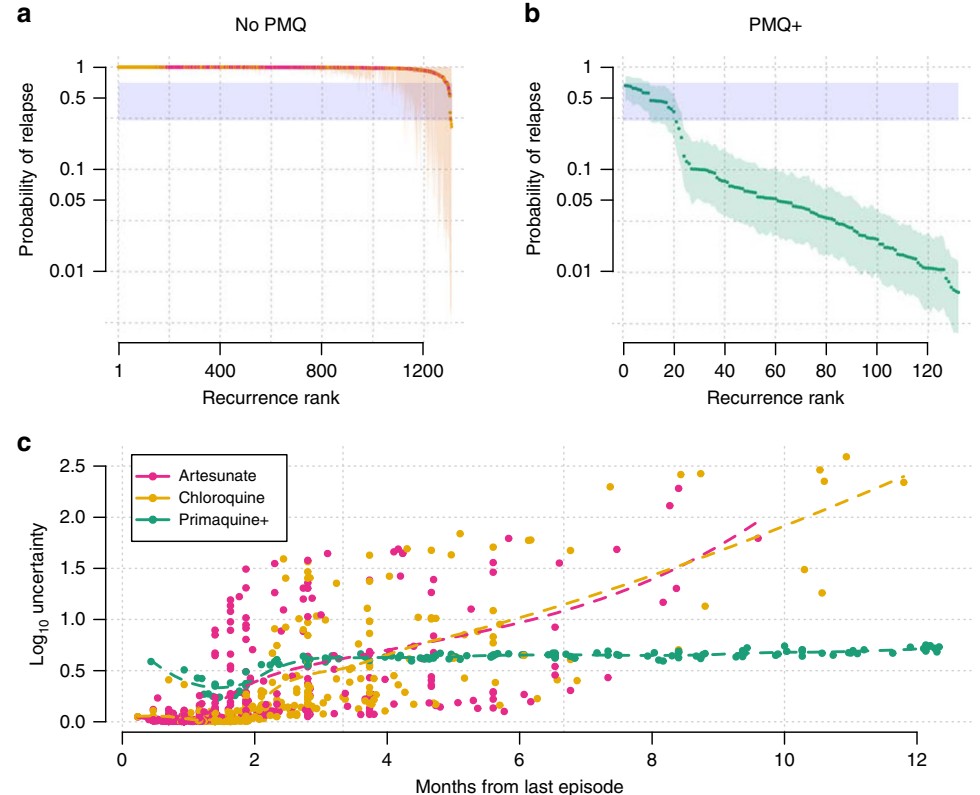

**Fig. 2 Probabilities of *P. vivax* relapse estimated under the time-to-event model.** Top panels: per-recurrence mean probability of relapse together with 95% credible intervals for 1309 recurrences following either artesunate or chloroquine monotherapies (No PMQ, **a**) and for 130 recurrences following high-dose primaquine with partner drug (PMQ+, **b**). The recurrences are ranked by their mean probabilities of relapse. The zone of uncertainty (same as in Fig. 3) is highlighted in blue. The upper and lower bounds are arbitrary. **c** The relationship between time since last episode and the uncertainty of the posterior estimates (width of the 95% credible interval on the $\log_{10}$ scale), coloured by treatment received. The dashed lines represent fitted LOESS curves to highlight trends in the No PMQ and PMQ+ groups, respectively.

**Table 2 Summary of *P. vivax* recurrences with probabilistic estimates of recurrence states derived from the time-to-event model.**

| Treatment | Recurrences (patient follow-up years) | Relapse[a] | Recrudescence[a] | Reinfection[a] |
|---|---|---|---|---|
| Artesunate monotherapy | 722 (161) | 96.1 (94.3–97.3) | 0.3 (0.2–0.5) | 3.6 (2.4–5.5) |
| Chloroquine monotherapy | 587 (169) | 95.0 (92.9–96.6) | 0.2 (0.1–0.4) | 4.8 (3.3–6.9) |
| *Primaquine+[b] | 132 (677) | 12.2 (9.3–15.9) | 0.01 (0.01–0.02) | 87.7 (84.1–90.7) |
| VHX: *Primaquine+ | 40 (155) | 9.2 (6.4–12.2) | 0.03 (0.01–0.06) | 90.8 (87.8–93.6) |
| BPD: *Primaquine+ | 92 (522) | 13.5 (10.1–17.3) | 0.01 (0–0.01) | 86.5 (82.7–89.9) |

*Primaquine+ refers to the radical cure combination treatment of primaquine together with one of two blood-stage treatments: chloroquine or dihydroartemisinin-piperaquine
[a]Probabilistic breakdown of recurrences: mean probabilities (95% CI) estimated under the time-to-event model. All estimates have been rounded up to one decimal place and are expressed as percentages
[b]VHX and BPD studies combined

25% had mean probabilities greater than 0.1 of being relapses and the remaining 75% had mean probabilities less than 0.1 of being relapses (Fig. 2**b**). In both cases, the time-to-recurrence is correlated with the posterior uncertainty with much lower uncertainty surrounding the posterior probabilities for recurrences following high-dose primaquine (Fig. 2**c**). The least uncertainty was observed around the peak expected timing of relapse following treatment. This interval to relapse is dependent on whether the blood-stage drug administered was slowly eliminated (chloroquine or piperaquine) or rapidly eliminated (artesunate), irrespective of whether high-dose primaquine was added.

The mean estimates of the recurrence states averaged over all observed recurrences in the pooled time-to-event analysis are given in Table 2. Considering mean probabilities, after supervised

high-dose primaquine in this epidemiological context, the model estimated that $\geq 84\%$ of the recurrences are reinfections, as compared to less than 7% when radical cure is not given. There was little evidence of recrudescence for all treatments considered. These results are consistent with previous modelling estimates from the same area[20]. In addition, the model estimated that the reinfection rate decreased by 53% (95% CI 37–64%) between the VHX study (2010–2012) and the BPD study (2012–2014) (Supplementary Table 1). This corresponds to a mean time to reinfection of 2.9 years (95% CI 2.4–3.6) during the VHX study and 6.1 years (95% CI 5.0–7.6) during the BPD study.

**Genetically informed estimates of recurrence states.** Evidence of sibling and clonal parasites across vivax episodes is strongly suggestive of either relapse or recrudescence (if clonal). A total of

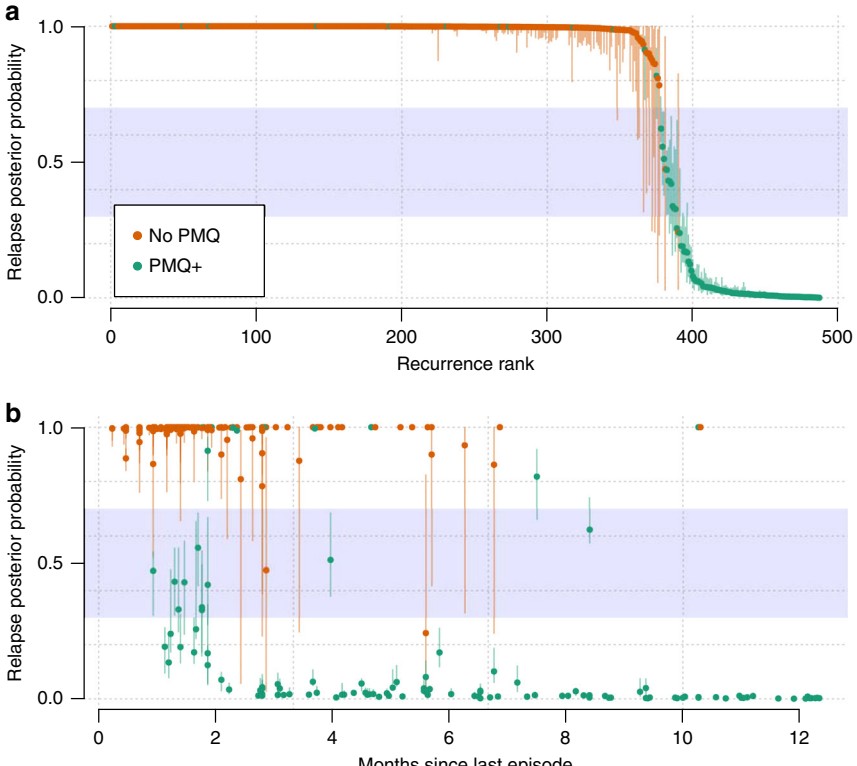

**Fig. 3 Probabilities of relapse for 487 genotyped *P. vivax* recurrences. a** All recurrences ranked by their probabilities of relapse coloured by treatment drug (orange: blood-stage treatment only, No PMQ; green: high-dose primaquine plus partner drug, PMQ+). In all, 95% credible intervals are shown by the vertical lines. **b** The same posterior probabilities as a function of the time since the last episode of *P. vivax* with the same colour coding. The uncertainty zone (same as in Fig. 2, and used to classify recurrences in Fig. 4) is shown by the blue zone (the upper and lower bounds are arbitrary).

710 episodes (of which 494 were recurrent) were genotyped from 217 individuals. Under the genetic model with a time-to-event based prior, we estimated recurrence state probabilities for 487 of the 494 recurrent episodes (enrolment data were missing for one and computational complexity under the genetic model prevented analysis of six), taking into account all available information. We estimated that in individuals who did not receive high-dose primaquine (all in VHX), nearly all (99.1%, 95% CI: 96.0–99.9) of the typed recurrences were relapses ($n = 366$). In contrast, for individuals who were given high-dose primaquine, only 10.8% (95% CI: 8.8–13.3) in the VHX study ($n = 34$) and 19.4% (16.6–23.7) in the BPD study ($n = 87$) of typed recurrences were estimated to be relapses. The estimates for recrudescence were very low: 0% (0–0) and 0.3% (0.2–0.5) in individuals who received high-dose primaquine and in those who did not, respectively. Overall, the vast majority of recurrent episodes for which we had genetic data had low uncertainty in the probabilities of their recurrence state (Fig. 3, vertical lines). We note that trial summaries based on probabilities of the individuals who did not receive high-dose primaquine are biased due to a preferential genotyping of the chloroquine-treated individuals with the highest number of recurrences (all in the VHX study). Of particular interest are two recurrences that occurred 300 days after an infection free interval with very high and certain probability of relapse (Fig. 3b), and thus were classified as failures (Fig. 4). These relapses occurred in two separate patients: one individual had received high-dose primaquine and the other had not.

We estimated the false-failure discovery rate of the genetic model by comparing data from episodes in separate individuals (null genetic data). Since relapses and recrudescences can only occur within an individual, any failure inferred in episodes from separate individuals is false. This gave an estimated false-failure discovery rate of 2.5% across in 249,540 pairwise comparisons, highlighting the discriminatory power of our panel of nine microsatellites (equivalent to a panel of approximately 50 biallelic SNPs; see Eq. (2)). This low rate also highlights the considerable population diversity of *P. vivax* within this small geographic area where transmission of malaria is low and seasonal.

**Reinfection-adjusted failure rate of high-dose primaquine.** We estimated the reinfection-adjusted failure rate after supervised high-dose primaquine to be 3.0% (95% CI: 2.4–4.0) in the BPD study, 2.4% (1.7–3.3) in the VHX study, and 2.9% (2.3–3.8) in both studies combined. Of 853 patients who received supervised high-dose primaquine, with 677 patient follow-up years (VHX and BPD combined), an estimated 2.5% (2.1–3.1) had at least one failure during follow-up. This estimate (2.5%) is slightly lower than the overall failure-rate estimate (2.9%) that accounts for loss to follow-up. In comparison, of 446 patients who did not receive high-dose primaquine with 330 patient follow-up years, 73.8% (62.4–76.6) had at least one failure during follow-up.

These estimates are based on the combined genetic and time-to-event model. For the BPD study, which contributed most of the data, the reinfection-adjusted estimate (3.0%) is significantly lower than the original reinfection-unadjusted estimate of the failure rate at 12% (95% CI: 10–14)[19]. A breakdown of how the time-to-event model and genetic model contribute to this final estimate is given in Supplementary Table 2. If we assume that the average hypnozoite burden was the same in the VHX and BPD patients, the mean recurrence count per patient follow-up year suggests that the relapse rate was reduced from 3.3 per patient follow-up year following chloroquine monotherapy to 0.04 per patient follow-up year following high-dose primaquine (Table 2).

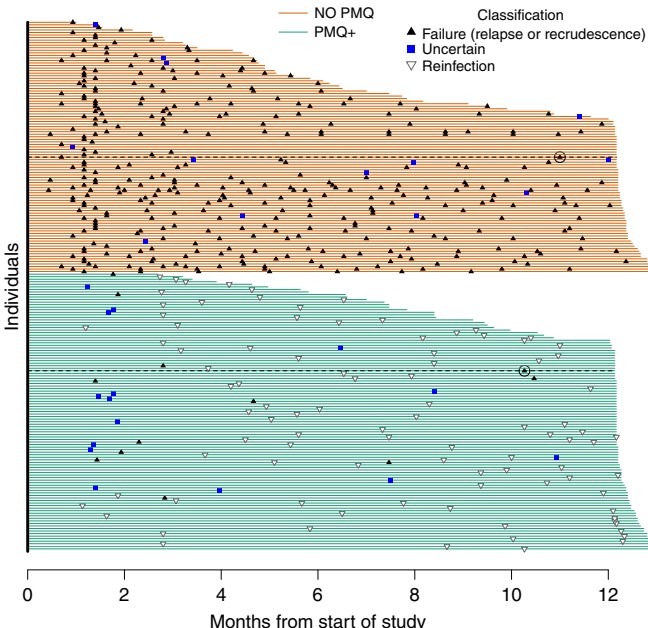

**Fig. 4 Classification of 487 genotyped *P. vivax* recurrences.** Each line represents one individual (*n* = 208). Duration of active follow-up is shown by the span of the horizontal lines (green: high-dose primaquine given, PMQ+; orange: no primaquine given, No PMQ). Recurrences classified as failures are black triangles (point up), reinfections are hollow triangles (point down), recurrences with uncertain classification are blue squares since they fall in the blue zone of uncertainty in Fig. 3. Since there was little evidence of recrudescence, all failures are essentially relapses. The delayed failures (circa ten months after treatment of the previous episode) are circled with their follow-up duration shown by a black dotted line. A histogram summary of these recurrence classifications is shown in Supplementary Fig. 1.

This suggests that 99% of microscopy detectable relapses were averted. The estimate of the number of relapses averted is not generalisable to other settings as it is a function of the individual hypnozoite loads and therefore of the background transmission intensity[3]. In fact, it could be a slight overestimate for the Thailand–Myanmar border as the transmission rate fell by half between the two studies.

To investigate these results further, we assessed the contribution of individual patient drug exposures by examining the relationship between the day 7 trough concentrations of carboxy-primaquine (the slowly eliminated inactive metabolite of primaquine) and treatment failure, adjusted for primaquine regimen administered (either 14 daily doses of 0.5 mg/kg or seven daily doses of 1 mg/kg). A statistically significant trend was observed, but this was driven by a few outliers, defined as episodes in which the plasma carboxy-primaquine trough concentrations were more than 3 standard deviations below the mean (Supplementary Fig. 2). Concentrations this far below expected values are likely to reflect incomplete drug absorption resulting from protocol deviations (e.g. non-adherence, vomiting). After removing these outliers, there was no statistically significant relationship between drug exposure and radical cure failure. This result illustrates the importance of discriminating between drug failures due to biological mechanisms (e.g. high hypnozoite load, cytochrome P450 2D6 polymorphisms, intrinsic drug resistance, etc.) and drug failures because of vomiting the medication or non-adherence. This is necessary for correct estimation of drug efficacy. Given the very low failure rate of supervised high-dose primaquine (estimated at 2.9%), only very

large pooled patient data analyses would have the necessary power to confirm or refute the conjecture that carboxy-primaquine concentration correlates with supervised high-dose primaquine treatment failure.

**Microsatellite requirements for recurrence state inference**. To inform microsatellite genotyping in future studies, it is important to determine the minimum number of markers necessary for reliable inference of the unknown recurrence states. Data on 3–12 independent microsatellite markers were simulated for paired infections (one primary episode followed by a single recurrence) under three scenarios: the recurrence contains a haploid parasite genotype that is either a sibling, stranger or clone of a haploid parasite genotype in the primary infection. To emphasise clearly the information content for a given number of markers we analysed the simulated data using a uniform distribution over the recurrence states (i.e. recrudescence, reinfection, and relapse each have prior probability of one third).

For each of the three scenarios, Fig. 5 shows the posterior probabilities of the recurrence states as a function of the number of markers simulated, assuming an effective cardinality of 13 per marker (the average effective cardinality in our panel of nine microsatellites, see Methods) and complexities of infection (COIs) of one in both the primary and recurrent episode. Under the stranger and clonal scenario, six or more markers sufficed to recover expected probabilities; see caption Fig. 5. Under the sibling scenario (exclusive evidence of relapse) nine markers or more were needed to obtain a median posterior probability of relapse close to one. Additional markers improve relapse and reinfection inference when the effective cardinality is low (Supplementary Fig. 3(i)), when there are errors in the genotyping (Supplementary Fig. 3(ii)), and when COIs exceed one (Supplementary Fig. 4). These simple simulations suggest that reliable posterior estimates of unknown recurrence states can be obtained with approximately nine or more highly polyallelic markers.

## Discussion

Distinguishing relapse, recrudescence, and reinfection in recurrent vivax malaria is necessary for the correct interpretation of therapeutic efficacy studies and for the optimal planning of malaria control and elimination. Our model framework is, to our knowledge, the first to generate individual probabilities of recurrence states using both time-to-event and genetic data with a model framework that incorporates IBD and multi-locus genetic data. The most operationally relevant result from applying this modelling approach to large clinical studies conducted on the Thailand–Myanmar border was a significant downward revision in the estimated radical curative failure rate. The reinfection-adjusted failure rate of supervised high-dose primaquine was estimated at 2.9% compared with reinfection-unadjusted estimates as high as 12%[19]. Approximately three in four patients in the VHX study randomised to no primaquine (*n* = 446) had at least one relapse during the follow-up periods. This contrasts with approximately 1 in 40 patients given high-dose primaquine (*n* = 853) who relapsed during the follow-up periods. The 30-fold decrease in the risk of relapse following supervised high-dose primaquine holds for recurrences detectable by microscopy and so could overestimate radical curative efficacy if there were sub-microscopic relapses. In any case, the radical curative efficacy of primaquine is not a fixed property even after reinfection adjustment (it depends on background transmission intensity and resulting hypnozoite burdens[3,21]), so this result reflects the value of both high-dose primaquine and effective malaria control in the area[22]. Moreover, it provides a benchmark for the development of

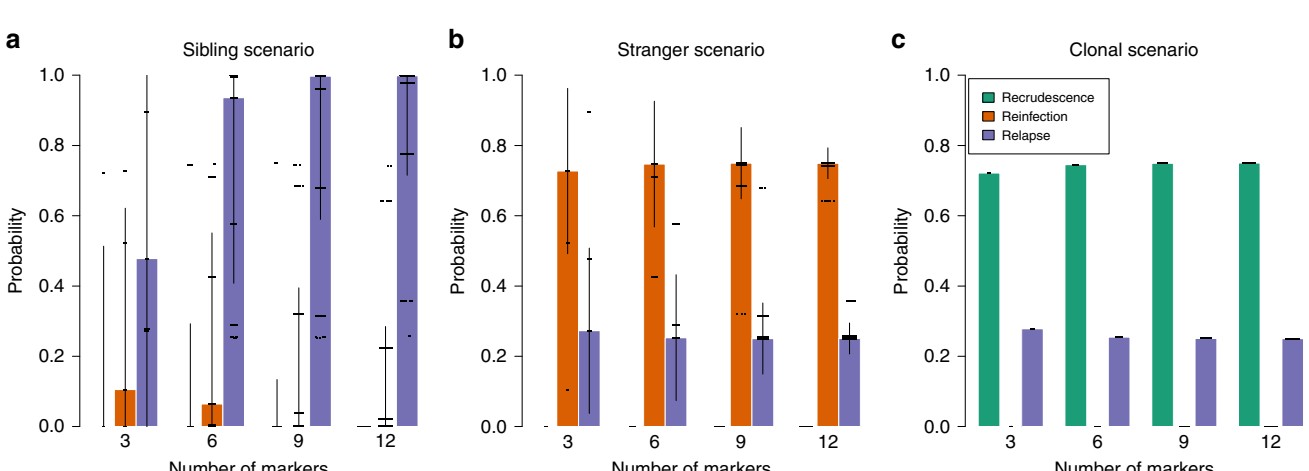

**Fig. 5 Probabilities of *P. vivax* recurrence states with simulated marker count.** Each plot corresponds to a different relationship simulation scenario: **a** sibling scenario; **b** stranger scenario; and **c** clonal scenario. Each coloured bar shows the median of 250 posterior probabilities (dots) with error bars extending ± two standard deviations. The effective cardinality of each marker was set equal to 13 in these simulations. The complexity of infection was one in both first and second infection. The prior probability of each recurrence state was $\frac{1}{3}$. As the number of microsatellites genotyped is increased, the following are expected: under the sibling scenario (**a**), the probabilities should converge to 1 for relapse and 0 otherwise; under the stranger scenario (**b**), the probability of reinfection should converge to a probability greater than the prior and the complement of that of relapse, meanwhile the probability of recrudescence should converge to 0; under the clonal scenario (**c**), the probability of recrudescence should converge to a probability greater than the prior and the complement of that of relapse, meanwhile the probability of reinfection should converge to 0.

new hypnozonticidal drugs, notably tafenoquine[23,24]. The benefit of single-dose tafenoquine, which solves the problem of adherence, needs to be balanced against its lower efficacy in Southeast Asia[23], and the increased proportion of individuals unable to take the drug because of the greater danger of haemolysis in heterozygous G6PD-deficient females[25], both of which are dependent on geographic context.

Much remains to be learned regarding the biology of relapse. No biomakers or good in vitro models are yet available. Under our time-to-event model we recovered an approximate 60:40 split between early/periodic relapse and late/constant-rate relapse. The 60:40 split is based on data from active follow-up and treatment of all cases (asymptomatic recurrences included)[18,19] on the Thailand–Myanmar border, where *P. vivax* exhibits the short-latency phenotype[3,4]. Our analysis takes into account post treatment prophylaxis from the slowly eliminated antimalarials and shows that, in this setting, the pattern of relapsing infections does not fit a simple constant-rate model (adopted elsewhere[20,21,26,27]). Nevertheless, a significant proportion of relapses do appear to arise at random (i.e. without periodicity), thus meriting the inclusion of a constant-rate compartment in the mixture model. Notably we observed two late recurrences (>300 days post previous infection) both with high and certain probabilities of relapse based on all available data. These results suggest that either some intermediate submicroscopic episodes were missed or some short-latency hypnozoites remained dormant in the liver for up to year, in agreement with previous reports[2,13]. These late relapsing hypnozoites likely awaken via a different mechanism to that of the highly periodic long-latency *P. vivax*[3]; the most parsimonious explanation would be that they awake at random.

Our model framework allows complementary information from different data types to be quantified systematically. However, strong assumptions are necessary and each model has its limitations. For example, we do not account for seasonality under the time-to-event model. Simulation suggests that this omission has little impact, but elsewhere it may have more bearing. A potential weakness in both the time-to-event and the genetic models is that neither allow for overlap of recurrence states. Individuals with large hypnozoite burdens who relapse frequently could be reinfected while they experience relapse. Both models would, on average, tend to label such an event as a relapse. Therefore, relapses can hide reinfection events. The main limitations of the genetic model are poor ability to infer a recrudescence and computational complexity. In general, correct classification of recrudescence is difficult because, when resistance is low-grade, recrudescent infections will reach patency at similar times to relapsing infections[3] and the genetic signature is the same as an infection arising from clonal hypnozoite(s). Our genetic model is also brittle with respect to recrudescence as we do not account for imperfect detection nor for genotyping errors; see Supplementary Figs. 3 and 4. As other indicators of reduced susceptibility (slowing of *P. vivax* parasite clearance rates and reduced in vitro susceptibility) do not suggest clinically significant blood-stage treatment failure rates in this area, this limitation is unlikely to have affected our results significantly, but it would necessitate modification before application to data from a region where there is significant *P. vivax* antimalarial drug resistance. The computational complexity increases with the number of recurrences and their COIs; it also hampers recrudescence inference (Supplementary Fig. 4(ii)). Since genetic complexity and diversity co-vary with transmission[28], our genetic model implies a sweet spot for inference where complexity is sufficiently low yet diversity sufficiently high that genetic data are informative. On a population level, *P. vivax* has high levels of genetic diversity even in low transmission settings[3,29–31], and the majority of *P. vivax* endemic areas do have low transmission[1]. However, the method would require modification before application to data from a high transmission setting such as Papua New Guinea[32,33].

Both Ross et al.[32] and White et al.[33] have modelled vivax recurrence data from Papua New Guinea. The model of Ross et al. was based on the presence–absence patterns of alleles at two polyallelic markers. Each marker was modelled separately, allowing for various complexities (e.g. imperfect detection that

varies over time). The model did not include either genetic recombination or the possibility that an individual is reinfected with parasites that have the same allele as parasites from a previous inoculation[32]. White et al.[33] combined time-to-event data with data from a single polyallelic locus under a model of parasite clone acquisition and clearance. Individual probabilities of relapse were estimated but were not fully identifiable and data on a single locus did not discriminate between a single blood-stage infection and multiple successive relapses (the majority of recurrent episodes were asymptomatic and were not treated). White et al. proposed that multi-locus genetic data would improve resolution of recurrence states. Our simple simulation study suggests that satisfactory inference of recurrence states can be achieved with nine or more highly polymorphic microsatellite markers. However, investment in obtaining more marker rich data (e.g. microhaplotypes[34,35]) is merited. There are no commensurable models of vivax recurrence with which to compare this result, but it agrees roughly with estimates from parentage and sibship studies[36,37]. The estimated marker requirements for informed recurrent state inference are modest, but the VHX and BPD data that enabled this approach are unusually comprehensive.

The overall requirements for the application of our model are summarised as follows. The time-to-event model relies on a good understanding of the *P. vivax* phenotype in context, e.g. is the phenotype largely of the early frequent relapse phenotype or are long-latency phenotypes prevalent? Long latency would require some minor modifications. The time-to-event model also relies on understanding any temporal changes to transmission dynamics, as in the data analysed here; and on characterising the pharmacokinetic properties of the blood-stage drugs (i.e. the terminal elimination half-life, which determines the continued suppression of parasite multiplication). This background knowledge must be encoded using strongly informative priors for identifiability of the parameters. In addition to prior understanding, active follow-up of a large number of individuals for several months is required (over 1000 patient follow-up years were available to support the current study). The genetic model also requires active follow-up with diagnosis and treatment of asymptomatic infections as it currently assumes all parasites are detected. Since correct classification of recrudescence is difficult and not robust under the current model, we recommend modification before application to a setting where there is significant *P. vivax* resistance. This necessitates prior knowledge of resistance in context. Due to computational complexity, the genetic model requires low COIs (ideally less than 4). At present, the genetic model would require modification before application to data from many markers, because of the computational complexity. Ideally, the genetic data should also derive from a diverse *P. vivax* population that supports genetic markers with high effective cardinality (the nine markers used in the current study equate to approximately 50 biallelic SNPs, Eq. (2)). However, the combined approach would not fail given entirely monomorphic genetic data: it would simply return an estimate based on time-to-event data. Most importantly, at least two episodes per person are required: the framework is designed to estimate recurrent states; it cannot estimate states for stand-alone episodes. As such, data from a cross-sectional study are not supported by this approach. The current framework (i.e. time-to-event plus genetic model that accounts for chance sharing of common alleles using an IBD-based approach) could be simplified and adapted for model-based distinction of recrudescence versus reinfection following treatment of *P. falciparum* in clinical trials that currently use a fixed time interval and IBS-based approach[7].

Models based on time-to-event data fit well generally to vivax malaria recurrences and can provide probabilistic estimates of the cause of recurrence. However, they necessitate large datasets and do not use important information regarding relatedness as captured by genetic data. On the other hand, genetic data alone cannot resolve relapse and reinfection when evidence of relatedness is lacking. Apart, each model is useful but sub-optimal. In combination they provide informed probabilistic estimates. Using a combined approach we determined that the radical curative efficacy of supervised high-dose primaquine is considerably higher than previously estimated in the epidemiological setting of frequent relapse vivax malaria on the Thailand–Myanmar border. This provides a comprehensive framework for resolving the cause of malaria recurrences and thereby contributes to an improved understanding of the biology, epidemiology, and treatment of *P. vivax* malaria.

## Methods

**Clinical procedures.** Both the VHX and BPD trials were conducted by the Shoklo Malaria Research Unit in clinics along the Thailand–Myanmar border in north-western Thailand, an area with low seasonal malaria transmission[18,19]. The patient populations include migrant workers and displaced persons of Burman and Karen ethnicity[38]. During the time these studies were conducted, primaquine radical cure treatment was not routine.

In both studies, recurrent episodes were detected actively at the scheduled visits by microscopy (lower limit of detection is approximately 50 parasites per μL). Patients were encouraged to come to the clinics between scheduled visits when unwell and so some recurrences were detected passively (less than 5%). All recurrences were treated, irrespective of symptoms.

**Ethical approval.** The BPD study was approved by both the Mahidol University Faculty of Tropical Medicine Ethics Committee (MUTM 2011-043, TMEC 11-008) and the Oxford Tropical Research Ethics Committee (OXTREC 17-11) and was registered at ClinicalTrials.gov (NCT01640574). The VHX study was given ethical approval by the Mahidol University Faculty of Tropical Medicine Ethics Committee (MUTM 2010-006) and the Oxford Tropical Research Ethics Committee (OXTREC 04-10) and was registered at ClinicalTrials.gov (NCT01074905).

**Vivax History trial (VHX).** This randomised controlled trial was conducted between May 2010 and October 2012. In total, 644 patients older than 6 months and weighing more than 7 kg with microscopy confirmed uncomplicated *P. vivax* mono-species infection (*P. vivax* only) were randomised to receive artesunate (2 mg/kg per day for 5 days), chloroquine (25 mg base per kg divided over 3 days: 10, 10, and 5 mg/kg), or chloroquine plus primaquine (0.5 mg base per kg per day for 14 days).

G6PD-deficient patients (as determined by the fluorescent spot test) were randomised only to the artesunate and chloroquine monotherapy groups. Subjects were followed daily for supervised drug treatment. Follow-up continued weekly for 8 weeks and then every 4 weeks for a total of 1 year. Patients with microscopy confirmed *P. vivax* infections were retreated with the same study drug as in the original allocation. Patients in the artesunate or chloroquine monotherapy groups who experienced more than 9 recurrences were given radical curative treatment with the standard primaquine regimen (0.5 mg base per kg per day for 14 days).

**Best Primaquine Dose trial (BPD).** Between February 2012 and July 2014, 680 patients older than 6 months were enrolled in a four-way randomised controlled trial simultaneously comparing two regimens of primaquine (0.5 mg/kg per day for 14 days or 1 mg/kg per day for 7 days) combined with one of two blood-stage treatments: chloroquine (25 mg base per kg) or dihydroartemisinin-piperaquine (dihydroartemisinin 7 and piperaquine 55 mg/kg). All doses were supervised.

The inclusion and exclusion criteria for this study were the same as for the VHX trial, except for the following: patients were excluded if they were G6PD deficient by the fluorescent spot test, had a haematocrit less than 25%, or had received a blood transfusion within 3 months.

Follow-up visits occurred on weeks 2 and 4, and then every 4 weeks for a total of one year. Any recurrent *P. vivax* infections detected by microscopy (same criteria as for VHX) were treated with a standard regimen of chloroquine (25 mg base per kg over 3 days) and primaquine (0.5 mg base per kg per day for 14 days).

**Microsatellite genotyping.** Whole blood for complete blood count was collected by venipuncture in a 2 mL EDTA tube. The remaining whole blood was frozen at −80 °C. *P. vivax* genomic DNA was extracted from 1 mL of venous blood using an automated DNA extraction system QIAsymphony SP (Qiagen, Germany) and QIAsymphony DSP DNA mini kit (Qiagen, Germany) according to the manufacturer's instructions. In order to compare the genotypic patterns of primary infections and recurrences, we genotyped initially using three polymorphic microsatellite loci that provided very clean amplification: no stutter peaks, and

reliability of PCR amplification at the low parasite densities usually found in recurrent infections. These core loci were PV.3.27, PV.3.502, and PV.ms8. A seminested PCR approach was adopted for all the fragments[12,39]. All amplification reactions were performed in a total volume of 10 μL and in the presence of 10 mmol/L Tris-HCl (pH 8.3), 50 mmol/L KCl, 250 nmol/L of each oligonucleotide primer, 2.5 mmol/L MgCl$_2$, 125 μmol/L of each of the four deoxynucleoside triphosphates, and 0.4 U of TaKaRa polymerase (TaKaRa BIO). Primary amplification reactions were initiated with 2 μL of the template genomic DNA prepared from the blood samples, and 1 μL of the product of these reactions was used to initiate the secondary amplification reactions. The cycling parameters for PCR were as follows: initial denaturation for 5 min at 95 °C preceded annealing performed for 30 s at 52 °C, extension performed for 30 s at 72 °C, and denaturation performed for 30 s at 94 °C. After a final annealing step was performed, followed by 2 min of extension, the reaction was stopped. PCR products were stored at 4 °C until analysis.

The genotypes of parasites in recurrent samples were compared with those in enrolment samples, and sample pairs were assigned a crude classification based on IBS, defined as related based on majority IBS, if two or three of three loci typed showed evidence of IBS, and different based on majority not IBS, otherwise. Heteroallelic calls had evidence of IBS if they included a call that was identical to another across the comparison. If the paired samples were classified as related based on majority IBS, or if one or more of the initial loci failed to amplify, six additional (non-core) microsatellite markers were genotyped (PV.1.501, PV.ms1, PV.ms5, PV.ms6, PV.ms7, and PV.ms16). For each microsatellite, details including the motif, chromosome, and position are provided in Supplementary Table 3. Counts of episodes partitioned by the number of additional markers typed successfully are provided in Supplementary Table 4. To see if additional markers bias relapse inference, we partitioned the probability of relapse inferred in the null genetic data by the number of markers used to estimate the probability of relapse. Additional markers do not bias relapse inference: the probability of relapse decreases from the prior with one to three markers, stabilising around 0.25 thereafter (Supplementary Fig. 5).

For allele calling on the microsatellites, the lengths of the PCR products were measured in comparison to internal size standards (Genescan 500 LIZ) on an ABI 3100 Genetic analyzer (PE Applied Biosystems), using GENESCAN and GENOTYPER software (Applied Biosystems) to measure allele lengths and to quantify peak heights. Multiple alleles were called when there were multiple peaks per locus and where minor peaks were >33% of the height of the predominant allele. We included negative control samples (human DNA or no template) in each amplification run. A subset of the samples ($n = 10$) were analysed in triplicate to confirm the consistency of the results obtained. All pairs of primers were tested for specificity using genomic DNA from *P. falciparum* or humans.

**Time-to-event model of vivax malaria recurrence**. For recurrent *P. vivax* infections in the VHX and BPD studies, we developed and compared two Bayesian mixed-effects mixture models describing the time-to-event data conditional on the treatment drug administered. The first model (model 1) assumed 100% efficacy of high-dose primaquine with only reinfection possible after radical cure. The second model (model 2) allowed for relapse and recrudescence following high-dose primaquine. A full list of assumptions relating to both models can be found in Supplementary Table 5. Model 1 served as a base model to assess robustness. Model 2 was used as the final model and all reported estimates are derived from it. Notation was chosen so as to be consistent with the mathematical notation for the genetic model (see below). Note that in the model notation that follows $n$ is an index, whereas above it is used to denote counts. For each individual indexed by the subscript $n \in 1..N$, we record the time intervals (in days) between successive *P. vivax* episodes (the enrolment episode is denoted episode 0). The last time interval is right censored at the end of follow-up. The models assume no selection bias from loss to follow-up. For the $n$th individual, data concerning the time interval $t$ (the time between episode $t-1$ and episode $t$) is of the form $\boldsymbol{x}_n^{(t)} = \{D_n^t, Z_n^t, C_n^t, S_n\}$, where $D_n^t \in \{\text{AS}, \text{CQ}, \text{PMQ+}\}$ is the drug combination used to treat episode $t-1$ (AS: artesunate monotherapy; CQ: chloroquine monotherapy; PMQ+: primaquine plus blood-stage treatment), $Z_n^t$ is the time interval in days, $C_n^t \in \{0, 1\}$ denotes whether the interval was censored where 1 corresponds to a right censored observation (i.e. follow-up ended before the next recurrence was observed) and 0 corresponds to an observed recurrent infection, and $S_n$ denotes the study into which the patient was recruited (1: VHX, 2: BPD). In general, let $\boldsymbol{x}_n = \{\boldsymbol{x}_n^{(0)}, \ldots, \boldsymbol{x}_n^{(T)}\}$ denote all available time-to-event data for the $n$th individual. Few recurrences (eight) occurred in the first 8 weeks for patients randomised to the dihydroartemisinin-piperaquine arms of the BPD trial, so we modelled the postprophylactic period of piperaquine as identical to that of chloroquine (i.e. PMQ+ includes both chloroquine and dihydroartemisinin-piperaquine as blood-stage treatments). In reality, the elimination profiles and intrinsic activities are slightly different, with piperaquine providing slightly longer asexual stage suppression than chloroquine.

In both models, time-to-recurrence is modelled as a mixture of four distributions, with mixture weights depending on the treatment of the previous episode. The mixture distributions correspond to the different recurrence states. The four mixtures are: reinfection, given by an exponential distribution; early (periodic) relapse, given by a Weibull distribution with treatment drug-dependent

parameters; late (constant-rate) relapse, given by an exponential distribution; recrudescence, given by an exponential distribution. Model 2 specifies different mixing proportions for the reinfection component in the non-primaquine and primaquine groups, $p_n^{\text{AS}} = p_n^{\text{CQ}}$ and $p_n^{\text{PMQ+}}$, respectively. The mixing proportion between early/periodic and late/constant-rate relapse within the relapse component is the same across primaquine and non-primaquine groups.

The likelihood for model 2 is given as

$$Z_n^t \sim p_n^{D_n^t} \mathcal{E}(\lambda_{S_n}) \left(1 - p_n^{D_n^t}\right) \left\{ \left(1 - c^{D_n^t}\right) \left(q \mathcal{W}(\mu_{D_n^t}, k_{D_n^t}) + (1-q)\mathcal{E}(\gamma)\right) + c^{D_n^t} \mathcal{E}(\lambda_{\text{RC}}) \right\}, \quad (1)$$

where $p_n^{(\cdot)} \in (0, 1)$ is the individual and drug-specific mixture probability of reinfection (we set the prior to reflect our belief that $p_n^{\text{AS}} = p_n^{\text{CQ}} < p_n^{\text{PMQ+}}$) and $c^{(\cdot)} \in (0, 1)$ is the nested drug-specific mixture probability of recrudescence.

The likelihood for model 1 is the same except that $p_n^{\text{PMQ+}} = 1$ (only reinfection possible). $\mathcal{E}(\cdot)$ denotes the exponential distribution. In both models, $\lambda_{S_n}$ is the study specific reinfection rate. The relationship between $\lambda_1$ and $\lambda_2$ is parametrised as $\lambda_2 = \delta \lambda_1$ where priors are specified for $\lambda_1$ and $\delta$. $\delta$ specified the decrease in transmission between the VHX and BPD study periods. $\lambda_{\text{RC}}$ is the recrudescence rate (assumed drug independent). $c^{D_n^t}$ is a drug-dependent nested mixing proportion between recrudescence and relapse. The time to relapse is itself a mixture distribution where $q$ is the doubly nested mixing proportion between early (first component) and late (second component) relapses. This is a fixed proportion across all individuals. The late/constant-rate relapses are parameterised by the rate constant $\gamma$. The early relapses are assumed to be Weibull distributed, denoted $\mathcal{W}(\cdot, \cdot)$, with drug-dependent scale parameters $\mu_{D_n^t}$ and shape parameters $k_{D_n^t}$ whereby with $\mu_{\text{CQ}} = \mu_{\text{PMQ+}}$ and $k_{\text{CQ}} = k_{\text{PMQ+}}$.

The individual marginal probability of reinfection is given by $p_n^{D_n^t}$; the individual marginal probability of recrudescence is given by $\left[1 - p_n^{D_n^t}\right] c^{D_n^t}$; the individual marginal probability of relapse is given by $\left[1 - p_n^{D_n^t}\right]\left[1 - c^{D_n^t}\right]$.

We used informative prior distributions (Supplementary Table 1) to ensure identifiability of the mixture components. Information content in the data, over and above that specified in the prior, was examined visually using prior-to-posterior plots. The prior-to-posterior plot for model 2 is shown in Supplementary Fig. 6. Identifiability of parameters was determined by simulation. Fifty synthetic datasets were drawn from each of the data generating processes defined by models 1 and 2 and a modified version of model 2 which incorporated seasonal reinfection. The seasonal component was estimated from the empirical distribution of week of enrolment in the BPD and VHX studies. The models were then fit to these simulated datasets and estimated parameters were compared to simulation-truth parameters. Supplementary Fig. 7 shows the estimated PMQ+ failure rates (using model 2) versus the true failure rates for data generated under model 2 (well-specified model fit), and for data generated under the seasonal version of model 2 (mis-specified model fit), respectively. Seasonal reinfection results in slight overestimation of the failure rate. Posterior model checking was done by simulating 500 synthetic time-to-event datasets under the posterior predictive distribution of the final model fit. The number of recurrences per person-year for each treatment arm was chosen as summary statistics used to compute posterior predictive $p$ values (Supplementary Fig. 7).

The stan models output (i) Monte Carlo posterior distributions for all model parameters; (ii) posterior estimates of recurrence states for each time interval $\boldsymbol{x}_n^{(t)}$; (iii) log likelihood estimates of each posterior draw. For each model, we ran eight chains with $10^5$ iterations, thinning per 400 iterations and discarding half for burn-in. Convergence of MCMC chains was assessed using traceplots assessing mixing and agreement of the eight independent chains. All these analyses can be replicated with the online github repository.

**Allele frequencies and effective cardinality**. For each microsatellite genotyped, allele frequencies were estimated using all available genetic data from the enrolment episodes (137 VHX, 79 BPD) and a multinomial-Dirichlet model (Supplementary Fig. 8). For each marker the effective cardinality $n^*$, defined as the number of alleles that provide the same probability of identity by chance given equifrequent allele frequencies, was estimated as one over the sum of the allele frequencies squared[40]. From the effective cardinalities we can compute the number of hypothetical biallelic SNPs that the nine microsatellites equate to as follows:

$$\text{Hypothetical SNP count} = \sum_{m=1}^{M} \log_{n_{\text{SNP}}^*}(n_m^*), \quad (2)$$

where $m$ is the index over the $M = 9$ microsatellites and the logarithm is base $n_{\text{SNP}}^*$, the assumed average effective cardinality of a hypothetical SNP. This is 2 for an ideal SNP and approximately 1.5 for a realistic SNP[40].

**Genetic model**. The genetic model outputs the probability that a recurrent *P. vivax* episode in a given individual is a recrudescence, relapse, or reinfection with respect to previously observed episodes, given three inputs: (1) prior probabilities that the episode is a recrudescence, relapse, or reinfection (in this work they are based on

time-to-event data); (2) a set of population-level allele frequency estimates; (3) available genetic data for the observed episodes for the given individual each with at most nine polyallelic microsatellite markers. To propagate uncertainty in (1) and (2), we draw 100 Monte Carlo samples from the time-to-event model and from the posterior Dirichlet distributions over allele frequencies for each marker. The genetic model does not capture uncertainty due to variation in the number of genotyped markers as it is computationally prohibitive to do so at present. Nonetheless, the genetic model does not over interpret limited data: when genotyped markers are few it simply returns estimates close to the prior. The rest of this section gives an informal description of the model. A detailed description with a list of assumptions and the full mathematical specification is in the Supplementary Methods.

For a given individual, parasites within and across infections are considered to either be strangers, siblings, or clones in relation to one another (strangers refers to all parasites whose shared ancestry dates beyond a single mosquito). The set of inter-parasite relationships can be represented by a fully connected graph. Each vertex represents a haploid genotype, and each edge between genotypes is labelled as a sibling or a stranger when the genotypes are contained within the same infection, or as a clone, a sibling or a stranger when the genotypes are from different infections. For complex infections, the number of vertices is set equal to the COI, which is defined as the maximum number of alleles per marker observed.

The model assumes that relapses can occur for all inter-parasite relationships across infections (strangers, siblings, and clones), whereas reinfections occur only as strangers, and recrudescences only as clones. The key steps in the model are as follows. First, we calculate the probability of the genetic data given a labelled relationship graph. Second, we calculate the probability of the proposed graph given that the recurrence episode is a recrudescence, a relapse, and a reinfection. Third, we sum over all possible graphs. The set of labelled graphs includes all possible ways to phase the microsatellite data (i.e. attribute alleles to haploid genotypes in complex infections) as well as all viable relationships between haploid genotypes. For example, if genotype A is a clone of B and B is a clone of C, the only viable relationship between A and C is clonal.

The concept of relatedness (probability of IBD) features in the first step. However, the model does not estimate relatedness. Instead, it estimates the probability of observing the data given IBD multiplied by the probability of IBD conditional on a specified relationship (e.g. 0.5 for siblings in an outbred population). This calculation makes use of allele frequencies (shared common alleles are liable to be identical but not necessarily due to descent, while shared rare alleles are more likely IBD). We then sum over the two possible IBD scenarios (alleles are IBD or not) to obtain the probability of the observed data conditional on the specified relationship,

$$\mathbb{P}(\text{data} \mid \text{relationship}) = \mathbb{P}(\text{data} \mid \text{IBD}) \times \mathbb{P}(\text{IBD} \mid \text{relationship}) + \mathbb{P}(\text{data} \mid \text{not IBD}) \times \mathbb{P}(\text{not IBD} \mid \text{relationship}).$$

This is computed for all the pairwise relationships in the relationship graph (see Supplementary Methods for full details).

The computational complexity of the genetic model limits it to the joint analysis of three episodes (two recurrences) per patient (in our data this is the case for 158 patients). For each individual with more than two recurrences (54 patients), we estimated pairwise probabilities of recurrence states between episodes (using the above model) and constructed an adjacency matrix. Relapse probabilities were then defined as proportional to the maximum estimated probability of relapse with respect to all preceding episodes, and those of recrudescence with respect to the directly preceding episode. The probability of reinfection is the complement of the probability of relapse plus recrudescence. These probabilities were then re-weighted to sum to 1.

**Genetic simulation**. We used simulation to explore marker requirements for recurrent state inference. As described above, data on 3 to 12 independent microsatellite markers were simulated for paired infections (one primary episode followed by a single recurrence) under three scenarios: the recurrence contains a haploid parasite genotype that is either a sibling, stranger, or clone of a haploid parasite genotype in the primary infection. The simulated data were analysed assuming a uniform prior over the recurrence states (i.e. recrudescence, reinfection and relapse each have prior probabilities of one third). For each of the stranger, sibling and clonal scenarios, we simulated data for an initial and recurrent infection with respective COIs 1 & 1, 2 & 1, and 1 & 2, with and without error; and respective COIs 3 & 1, without error. To illustrate the behaviour of the model when applied to erroneous data, data with error were simulated using an extremely high per-locus probability of error (0.2 versus realistic error < 0.01[41]). When COIs exceeded one, the sibling, stranger, or clone was among unrelated stranger haploid genotypes (a relationship graph with at most a single non-stranger edge). For a given set of COIs, this type of graph yields highly diverse data and is thus the most challenging to analyse. For non-erroneous episodes with COIs in 1 or 2, we explored cardinalities of 13 and 4 (the average and minimum, respectively, of our panel of nine microsatellites). For the erroneous data and for the episodes with COIs of 3 & 1 we used cardinality equal to 13 only. The results of an illustrative subset of the genetic simulations are presented in Fig. 5 and Supplementary Figs. 3 and 4. All the genetic simulations can be replicated from the online github repository, see folder Simulation_Study.

**Classification of recurrent episodes**. The estimation of the false-failure discovery rate of the genetic model and Fig. 4 both necessitate the specification of classification boundaries. We arbitrarily chose the interval [0.3, 0.7] as the zone of uncertainty. Each recurrence is either classified as a reinfection or a failure where failure is either a relapse or a recrudescence: if the sum of the upper credible intervals of relapse plus recrudescence is less then 0.3, the recurrence is classified as a reinfection; if the sum of the lower credible intervals of relapse plus recrudescence exceeds 0.7, the recurrence is classified as a failure; otherwise classification is deemed uncertain. Since there was negligible evidence of recrudescence, all failures are essentially relapses.

**Reporting Summary**. Further information on research design is available in the Nature Research Reporting Summary linked to this article.

## Data availability
All deindentified microsatellite and time-to-event data that feature in this study can be found on the github repository (https://doi.org/10.5281/zenodo.3368828).

## Code availability
All model code and statistical data analysis algorithms were written in R (version 3.4.3). The genetic model uses the R package igraph[42]. Time-to-event models were written in rstan based on the stan probabilistic programming language[43,44]. Logistic and Poisson mixed-effects regression models were fitted using the package lme4. The complete models along with accompanying R code can be found on github at github.com/jwatowatson/RecurrentVivax/ (https://doi.org/10.5281/zenodo.3368828).

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

## Acknowledgements

N.J.W. is supported by a Principal Fellowship from the Wellcome Trust. K.P. is supported by the Royal Golden Jubilee Ph.D. Programme, the Thailand Research Fund (PHD/0032/2556). A.R.T. and C.O.B. were supported by a NIGMS Maximizing Investigator's Research Award (MIRA) R35GM124715-02. This project has been funded in part with Federal funds from the National Institute of Allergy and Infectious Diseases, National Institutes of Health, Department of Health and Human Services, under Grant Number U19AI110818 to the Broad Institute (to D.E.N.). The clinical studies were supported by a programme grant from the Wellcome Trust (reference 045143) and were part of the Wellcome Trust Mahidol -Oxford Tropical Medicine Research Programme. The content is solely the responsibility of the authors and does not necessarily represent the official views of the funders.

We are grateful to all the patients who took part in these studies and for the study staff who cared for them. A special thanks to Dr. Clare Ling, Dr. Germana Bancone, and Pornpimon Wilairisak for managing and keeping in order the large volume of study samples.

## Author contributions

Methodology, formal analysis, visualisation, and writing (original draft): A.R.T. and J.A.W. Resources and data curation: K.P., J.D., C.S.C. and M.I. Supervision: N.P.J.D., F.N., D.E.N., C.O.B., M.I. and N.J.W. Writing (review and editing): A.R.T., J.A.W., C.S.C., N.P.J.D., F.N., D.E.N., C.O.B., M.I. and N.J.W. Funding acquisition: D.E.N., C.O.B., M.I., N.P.J.D. and N.J.W. Conceptualisation: N.P.J.D. and N.J.W. All authors read, revised and approved the paper.

## Competing Interests

The authors declare no competing interests.
