## [Peer Review File · Nature Communications]

Reviewers' comments:

Reviewer #1 (Remarks to the Author):

This paper was quite annoying to read because it's essentially a much better version of a paper I recently published trying to do the same thing with inferior data. At the same time, it was a pleasure to read with some cool approaches tied together in an imaginative way. The degree of my comments below reflect my enthusiasm for this work rather than a desire to criticise.

Major comments

- Validation of methods on simulated data.

When utilising a newly developed computational method such as this, it's incredibly valuable to first demonstrate that it works on simulated data, e.g. simulate some data from the assumed model with a fixed set of parameters, and assess the ability of the inference method to identify them.

- Epidemiological overview.

A table providing an epidemiological overview of the data from the VHX and BPD clinical studies would be incredibly useful. Please include information in tabular form for: number of participants; duration of follow-up; treatment administered; covariates (age, gender, etc); clinical incidence; and anything else that help the reader understand the epidemiology of the study sites.

- Presentation of priors and posteriors.

Please present all parameter estimates, and the associated priors in an easy to read table (possibly in the appendix). I appreciate some of this data is already available in Figure S4, but it would be good to have it tabulated (e.g. posterior medians with 95% credible intervals), along with descriptions of the parameters.

- Parasite detection.

"In both studies, recurrent episodes were detected actively at the scheduled visits by microscopy (lower limit of detection is circa 20 parasites per μL)"

Systematic reviews have shown that a very large proportion of *P. vivax* infections are sub-microscopic. Can the authors comment on the robustness of the methods to undetected relapses.

Minor comments

- In Figure 1D there's a strong peak in relapse probability at 2 months (approximately 40%), with a corresponding reduction in re-infection probability. I wonder how strongly this pattern is driven by the underlying estimated Weibull distribution. A second point which the authors may wish to comment on is the possible role that 'bunching' may play, whereby blood-stage infections deriving from relapses are suppressed by prophylaxis and emerge after drug concentrations have waned.

- Data visualisation. The timing of recurrences are shown in Figure 4, but it's quite hard for the reader to mentally aggregate this data. Creating a histogram of this data, perhaps with classification would be quite helpful (possibly one for the Appendix).

- "The dynamics of time-to-relapse, irrespective of whether primaquine was administered, were explained as a 60/40 mixture between a highly periodic timing and a 'random' timing"

Please provide a table (perhaps in the Supp Appendix) with the parameter estimates and uncertainty corresponding to this statement.

- Sometimes it's unclear which estimate comes from which study: VHX, BPD, or joint.

- Figure 4: Why is one of black triangles for relapses circled for less than 1 month. Was there any classification of recrudescence?

- “Although recommended in most endemic countries, primaquine is not used widely because of the risks of iatrogenic haemolysis in patients with glucose-6-phosphate dehydrogenase (G6PD) deficiency”

This may be true in Asia, but not so in most of South America. In Brazil, for example, nearly 5 million people have been treated with primaquine without testing for G6PD deficiency since 2000.

- The last paragraph of the Introduction is a bit out of place and reads as a summary of the results.

- The bottom panels of Figure 2 would be better on the same scale on the y-axis.

- There seem to be a mix of 80% and 95% credible intervals reported. Please stick with the more conventional 95% intervals.

- The reinfection-adjusted estimates of primaquine efficacy are a particularly important contribution to the literature. Perhaps important enough to be noted in the abstract.

- “If we assume the background transmission to have been constant over the 4 years studied (2010-2014)”

Constant at what rate? This information could be included in the suggested table describing the epidemiology of the trial sites.

- For the Bayesian methods described in Section 4.3.1, were there 6 or 8 chains?

- Supplementary Figure 1: “The expected relatedness of meiotic siblings is the mean of a bimodal distribution with modes at 0.3 and 1 (see Appendix and 41,42 for explanation)”

Figure 2 of Wong et al gives modes of 0.33 and 1. Perhaps I’m missing something.

Reviewer #2 (Remarks to the Author):

Leveraging two large vivax treatment trials carried out on the Thai-Myanmar border that involved >1000 patients, Aimee Taylor and colleagues have constructed two relapse prediction models based on 1) time-to-event data from ~1600 recurrences and 2) microsatellite genotyping data from ~500 paired recurrences in ~200 persons. They then combine these two models into a joint Bayesian model whereby the time-to-event probabilities serve as the prior estimates for the genetic model. This combined model finds that high-dose primaquine is highly effective in the region for radical cure/preventing relapse, bringing the re-infection adjusted failure rate down to 2-3% from a previous estimate of 10-15%. This research addresses a persistent problem in the vivax field – measuring the efficacy of primaquine in endemic settings – in a novel way. The findings, if true, are important and would be of widespread interest.

I cannot follow all the math, but my understanding is as follows. In Model 1, predictions are based on a higher rate of relapse after short-acting drugs (artesunate, chloroquine) and a higher likelihood of re-infection further out, with different rates/distributions for relapse vs. reinfection. In Model 2, between 3-9 microsatellite markers are used to categorize episodes genetically as clonal, sibling, or stranger to the previous episode based on pair-wise relatedness that is meant to reflect

identity-by-descent (IBD) relatedness arising from meiosis of unrelated gametes in the mosquito. The model is driven by the key assumption that only relapses occur as siblings (but can be clonal or strangers), whereas re-infections occur only as strangers, and recrudescences are rare and clonal.

The overall scheme is clever in that it combines two disparate sources of predictive priors in a setting where many factors are relatively known – low transmission, lack of antimalarial resistance, intermediate complexity of infection, great genetic diversity despite low transmission, short relapsing strains, and common occurrence of heterologous relapses. The findings, that the vast majority of recurrences in SE Asia in the absence of primaquine treatment are due to relapse, is consistent with prior published estimates and expert opinion.

My primary question/concern is whether there is overinterpretation of limited genetic data leading to a false claim of accurate classification of IBD relationships. Another weakness that is less problematic is the uncertain applicability of the overall joint model to other areas where transmission is higher and antimalarial resistance rates are higher or unknown (this is acknowledged in the Discussion).

Major comments

1. IBD classification based on pairwise genetic relatedness.

Introduction - Top of page 4 "Relatedness is based on identity-by-descent, an approach that accounts for parasites that are genetically identical by chance, ensuring a low false positive rate of relapse classification (2.2%)."

This sentence was confusing to me in the beginning – I could not understand what a low false positive rate of relapse classification meant. Even after reading everything, I find the sentence misleading because I would not think that pairwise relatedness of 3-9 microsatellite markers is enough to ensure accurate IBD classification. If all genomic loci are considered, meiotic siblings would have pairwise relatedness = 0.5 (S2), but is the converse true when looking at a very small set of markers? – if 6 microsatellite markers covering only half the chromosomes (assume 3 of the 9 failed or were misclassified based on stutter peaks) had pairwise relatedness close to 0.5, does this mean the genotypes parasites are definitely siblings, and therefore relapse? In my understanding, the genetic model assumes that siblings are relapses rather than re-infection, but 0.5 pairwise relatedness at 6 repeat loci seems like it could represent re-infection as well. How would the model interpret a recurrence with MOI of 3, containing a clone and two unrelated strains, but with overall pairwise relatedness close to 0.5?

2. 4.2 Microsatellite genotyping. Since the number of markers differed for different samples, please report somewhere how many pre- and post-treatment genotypes were found to be different vs. related. For the related pairs, how many were successfully genotyped at 6 additional microsatellite loci? (since the modeling suggests that 9 markers are needed to accurately classify siblings). How many of the "related" pairs were ultimately classified as clones vs. siblings vs. strangers – or what was the distribution of pairwise relatedness?

If 10 samples were analysed in triplicate, how many were analyzed singly or in duplicate? If singly, then there is likely to be significant misclassification error due to slippage and stutter peaks.

3. Model details in Results. Since the novelty of the paper lies in the Bayesian models, the format of listing Methods in the last section doesn't serve the paper well, and I would've appreciated if more of the model design was incorporated into the Results section, rather than giving a brief review then reporting results which cannot be evaluated without diving into the supplementary material.

4. Time-to-event model. The assumptions in the genetic model and their implications are well laid out in the supplementary material. It would be nice to likewise list assumptions made in the time-to-event model.

Minor comments

5. 2.1 Overview of analysis – What were the “strongly informative priors for identifiability” of recrudescence and re-infections?
6. Fig 5 – “The complexities of infection were 1 in both first and second infections.” I thought elsewhere, the models accounts for COI up to 3. Many studies have shown that vivax infections in SE Asia are more often polyclonal than not. Why is COI set =1 here?
7. S7, p.31 – “4 ways two select two sporozoites” - typo

Reviewer #3 (Remarks to the Author):

For *P. falciparum* malaria, genotyping has been used to distinguish recrudescence and reinfection when assessing the efficacy of drugs aimed at blood-stage parasites. For *P. vivax*, dormant liver-stage parasites can lead to relapse. The possibility of relapse complicates the interpretation of drug trials since relapses may be clones or siblings of the blood-stage infection associated with the illness. This paper describes a method to combine information about the timing of recurrences and about genetic markers to provide estimates of the probability that a recurrence is a relapse, reinfection or recrudescence. This provides a more accurate picture of the efficacy of a liver-stage drug. The work is novel to my knowledge, and the paper is clearly and concisely written.

The estimates produced by this method are more sophisticated than just taking all recurrences together or by omitting only clear reinfections. It is plausible that the estimates are also more accurate, but this is not known and it may be that some assumptions are misleading (such as distributions being different to those assumed). There is some validation by simulation (section 2.5), but as far as I can work out, this assumes that all of the distributional assumptions are correct. How robust is the method if assumptions about, for example, seasonality are not correct? (Reinfection would usually be expected to be seasonal in a seasonal setting). Or if genotyping errors or imperfect detection of minority clones are incorporated?

The numbers in Table 1 from the time-to-event analysis suggest that on average the rates of reinfection in the groups are estimated to be around 10 per 100py for ART and CQ and around 17 per 100py for primaquine+. A crude simple test assuming a Poisson distribution suggests that this is significantly higher. It is not obvious why the reinfection rate should be higher in the PQ arm - is it possible that this is an artefact of the time-to-event model rather than a chance finding?

Participants have *P. vivax* illness at enrolment. It is not necessarily the case that illness only occurs following the primary infection, it may also occur as a consequence of relapse (from malaria therapy studies). The logic in the model seems to be that following enrolment subsequent relapses from the same inoculation follow a distribution, however the distribution may change depending on whether the illness visit was a consequence of a primary infection or a relapse.

A useful thing to know is what contribution the time information and the genetic information made in the final overall estimate. Figure S2 shows this for individual points, but how does each component change the overall estimates of drug efficacy? There seems to be a big shift from the priors in Figure S4.

As far as I can work out, the positions of the alleles on the chromosomes are not taken into account. For siblings, this could provide information in that breaks in chromosomes are more likely

between alleles further apart than close together.

Using the time-to-event data as a prior and the genetic data as the data should have a rationale. The justification on pS17 "it remains to be seen whether a formally joint model of both data types would add value" does not seem strong.

There are some statements scattered in the discussion and supplementary information about what type of data would need to be collected and where the method would work. It would be helpful to have one paragraph with these collected together to make it easy to understand the circumstances in which this method would work. For example, the timing of monitoring to get accurate estimates of time-to-recurrence, or the background parasite diversity.

Minor comments

Introduction. Last sentence of paragraph 1: "Reinfection rates will either be constant over time or seasonal". It is not obvious why they would be constant... there are very few sites with truly constant transmission.

The title talks of "resolving" the cause of recurrences but this claims too much, they cannot be resolved so much as have probabilities attached.

p8. Is it necessary to assume that the background transmission intensity was constant from 2010-2014? Could this not be estimated using either this data or another source?

p8. In section 2., it is stated that the reinfection-adjusted failure rate of primaquine is 2.6% (80% CI: 2.0-3.5), and this is compared to an estimate of 12% (80% CI 10-14). I could not work out if the 12% came from one of the two studies or both (only one is referenced). This makes it trickier to compare the two estimates.

p13. The relapses are assumed to come from a mixture distribution with one part occurring 'at random'. The term 'random' is not specific. A random variable can follow a distribution, it just must have a stochastic element. It is the underlying distribution that would be important here.

p17. Section 4.3.3 I can see that the authors have tried to explain the genetic model without statistics. However, I think it could be made simpler and more intuitive.

Could recrudescence plausibly happen at all 10 months after the clinical episode? (Figure 1). The probability is very low, but the decline might go to zero very quickly after a certain time interval.

How drug studies use time data (eg a fixed time window) and genetic data (eg for *P. falciparum*) currently could helpfully be mentioned in a sentence in the introduction.

Reviewer #1

This paper was quite annoying to read because it's essentially a much better version of a paper I recently published trying to do the same thing with inferior data. At the same time, it was a pleasure to read with some cool approaches tied together in an imaginative way. The degree of my comments below reflect my enthusiasm for this work rather than a desire to criticise.

We thank the reviewer for these kind comments and for their time taken over the manuscript review.

Major comments

- Validation of methods on simulated data.

When utilising a newly developed computational method such as this, it's incredibly valuable to first demonstrate that it works on simulated data, e.g. simulate some data from the assumed model with a fixed set of parameters, and assess the ability of the inference method to identify them.

We agree that this is extremely important, not only as a 'sanity check' (e.g. debugging of newly developed code) but also as a way of detecting potential weaknesses in the model. We apologize for not having included such simulation studies for the time-to-event model in the original submission. We had included, however, in the original submission, simulation scripts and results from simple simulations for the genetic model. For simplicity of presentation we had chosen not to give any results for complexities of infection greater than one.

Time-to-event simulations

In response to this comment, we have constructed in depth simulation scripts for the time-to-event model. These can be found at the github repository under the folder 'Simulation_Study/SimulationStudy_Timing_Model.Rmd'. In the paper we alert the reader to these simulation-based model checks in the penultimate paragraph of the updated Methods section 4.3.1.

In summary, the simulation scripts do the following:

- Simulate data under the assumptions of Model 1 and fit Model 1 to these data (well-specified problem). 100 random iterations showing true versus estimated parameters suggest no major apparent biases in parameter estimates.
- Simulate data under the assumptions of Model 2 and fit Model 2 to these data (well-specified). Again 100 iterations show no major apparent biases.
- Simulate data under the assumptions of Model 2 with an additional seasonal element whereby reinfection has a non-constant hazard rate as a function of the time of year. This fits Model 2 to these data (mis-specified). The 100 iterations evaluation shows a small bias in the estimation of the reinfection rate (underestimates the reinfection rate) and in the overall estimate of primaquine failure (overestimates treatment failure).
- Bayesian posterior model checks: simulate data under the assumptions of Model 2 with parameters drawn from the posterior distribution of Model 2 fitted to the pooled data set. We

then compare distributions of time-to-event for the three treatment arms between the simulated and observed data. We also compute posterior predictive p-values for the main summary statistics (the number of recurrences observed per patient) for each treatment arm.

All these simulations are written in a fully modular fashion, such that it is easy to extend and modify them. We believe that these results show that the overall model fitting process is not subject to major biases even though the model is clearly mis-specified (e.g. we know that reinfection is seasonal along the Thai-Myanmar border). The model parameters are identifiable under some assumptions and if strongly informative Bayesian priors are used. In addition, we believe that the strongly informative Bayesian priors are justified in view of the mechanistic understanding of relapse and recrudescence in vivax malaria. Supplementary Figure 7 summarises the main findings of this simulation study.

We thank the reviewer for suggesting these simulations for the time-to-event model. Whilst doing this it was noticed that the parameterization of the rate parameters for reinfection, recrudescence and late relapse lead to some instability. We have therefore changed the prior distributions for the rate parameters (before this was parametrized as a normal distribution over the inverse rate parameter) to the more standard choice of gamma distributions.

Genetic Simulations

We have extended the simulations under the genetic model. For each of the stranger, sibling and clonal scenarios, we simulated data for an initial and recurrent infection with respective complexities of infection (COIs): 1 and 1, with and without error; 2 and 1, with and without error; 1 and 2, with and without error; 3 and 1, without error. As before, data were simulated for 3 to 12 markers. For the non-erroneous episodes with COIs of 1 and 2, we explored cardinalities of 4 (the minimum effective cardinality of any marker in the MS data that features in the main text) and 13 (the mean effective cardinality of the MS data that feature in the main text). For the erroneous data and for the episodes with COIs of 3 and 1 we used cardinality = 13 only. These data and results can be found at the github repository under `Simulation_Study/SimulationStudy_Genetic_Model.Rmd` and respective pdf. We alert the reader to these simulations in section 2.5, which has been modified, and in a newly added section 4.3.4, entitled Genetic simulation. Illustrative results are presented in main text Figure 5, as before, and Supplementary Figures 3 and 4, which are new. More details are provided below.

SimulationStudy_Genetic_Model.Rmd relies on BuildSimData.R, a function which generates data for a relatedness graph over an initial episode and single recurrence, given a set of input parameters. The graph contains a single between-episode edge representing a parasite haploid genotype with specified relationship of stranger, sibling or clone, which is among otherwise unrelated stranger parasites if the input COIs exceed one. This type of graph (in which noisy parasite haploid genotypes are unrelated) is the most diverse and thus the most computationally challenging (an extreme scenario in comparison to those observed in the field data).

Results: As before, when the effective cardinality is as high as 13 (the mean effective cardinality of our MS data), nine markers provide sufficient information for reliable recurrent state inference providing the COIs of the initial and recurrent infections are both one. In the main text, we reference the following additional results. It is important to note that these results show the performance of the model under a prior that puts equal weight on each of the three recurrence states. However, in our analysis of the VHX and BPD data, the prior is based on informative time-to-event data.

1. When data are few the genetic model apportions posterior probability over potentially ambiguous states. Its confidence increases with the number of markers. This means that when data are few the genetic model is conservative and returns estimates close to the prior, guarding against over interpretation of limited data. When there are no data the genetic model returns the prior (PriorReturnCheck_Genetic_Model.R). This is a useful debugging test for any Bayesian model.
2. Additional markers improve relapse and reinfection inference when the effective cardinality is low (Supplementary Fig 3a), when the data are highly erroneous (Supplementary Fig 3b), and when the COIs exceed one (Supplementary Fig 4); see updated section 2.5. Specifically, three additional markers (now totalling twelve) improve inference when the COI of the initial or recurrent infection increases to two (Supplementary Fig 4a). When the COI of initial infection increases to three (Supplementary Fig 4b), a small fraction of simulations under the sibling scenario have non-zero reinfection probability using twelve markers, suggesting potential gain with more markers at higher levels of transmission intensity. To achieve zero reinfection probability under the sibling scenario when the data are very highly erroneous (e.g. if 20% of markers are typed incorrectly) more than 12 markers are required (Supplementary Fig 3b).
3. Under the model, clones are misinterpreted as siblings if the data are highly erroneous (genotyping error), thereby biasing recrudescence inference, but not the inference of relapse. Previously, this was noted in the discussion and in Supplementary section 3; Supplementary Fig 3b now illustrates it.
4. Supplementary Fig 4 now supports the pre-existing discussion around the frailty of recrudescence versus relapse inference in the presence of undetected minority clones; this is discussed in more detail in the caption of Supplementary Fig 4.
5. Increasing computational complexity biases inference of recrudescence: this is dependent on the number of markers typed and the complexities of infection. This problem is illustrated using highly diverse and thus computationally challenging simulated episodes with initial and recurrent COIs of 3 and 1, respectively (Supplementary Fig 4b), and described in the caption of Supplementary Fig 4b. It is not a problem in the real field data. In the field data, complex infections most likely derive from either relapsing or co-inoculated parasites (there is little opportunity for superinfection because of low transmission intensity and active follow up). As such, the parasites within infections are liable to be interrelated and thus less diverse. It would, however, be a problem if the model was used to analyze more markers. We thus refer briefly to this limitation in the discussion with reference to Supplementary Fig 4b.

- Epidemiological overview.

A table providing an epidemiological overview of the data from the VHX and BPD clinical studies would be incredibly useful. Please include information in tabular form for: number of participants; duration of follow-up; treatment administered; covariates (age, gender, etc); clinical incidence; and anything else that help the reader understand the epidemiology of the study sites.

We agree that this would provide useful information for the reader. Table 1 now gives a summary of the main enrolment and follow-up statistics for the two trials broken down by treatment assignment and

clinical study. We have also added information regarding the counts of individuals with episodes genotyped and recurrences genotyped.

- Presentation of priors and posteriors.

Please present all parameter estimates, and the associated priors in an easy to read table (possibly in the appendix). I appreciate some of this data is already available in Figure S4, but it would be good to have it tabulated (e.g. posterior medians with 95% credible intervals), along with descriptions of the parameters.

We thank the reviewer for this suggestion and agree that this should be included for completeness. We have added a supplementary table with this information (see Supplementary Table 1).

- Parasite detection.

“In both studies, recurrent episodes were detected actively at the scheduled visits by microscopy (lower limit of detection is circa 20 parasites per μL)”

Systematic reviews have shown that a very large proportion of *P. vivax* infections are sub-microscopic. Can the authors comment on the robustness of the methods to undetected relapses.

This is an important point that depends on the definition of ‘recurrence’. Here, the goal was to label probabilistically recurrences that were detectable by microscopy. The efficacy estimation of interest for high-dose primaquine pertains to microscopy detectable infections only (i.e. the most clinically relevant). To clarify, we have added the following statement to the first paragraph of the discussion: “The 30-fold decrease in the risk of relapse upon receipt of supervised high-dose primaquine holds for recurrences detectable by microscopy and so could overestimate radical curative efficacy if there were sub-microscopic relapses.”

Missing sub-microscopic infections could bias inference for microscopy detectable recurrent infections if sub-microscopic infections differ systematically in terms of their genetic signature and timing. This could happen theoretically if the immune response targeted genetically related relapses preferentially, as discussed in the ultimate paragraph of the Supplementary text. Our time-to-event model is purely descriptive and as such can only describe the time-to-recurrence for higher parasite density recurrences. These are the only clinically relevant events and they are also the only events proven to transmit vivax malaria.

Minor comments

- In Figure 1D there’s a strong peak in relapse probability at 2 months (approximately 40%), with a corresponding reduction in re-infection probability. I wonder how strongly this pattern is driven by the underlying estimated Weibull distribution. A second point which the authors may wish to comment on is the possible role that ‘bunching’ may play, whereby blood-stage infections deriving from relapses are suppressed by prophylaxis and emerge after drug concentrations have waned.

The peak in relapse probability at 2 months is in part driven by the Weibull distribution (accounting for approximately 60% of relapses), but is also driven by the estimated mixing parameter $p^{\text{PMQ}+}$ which

determines the probability of relapsing after high-dose primaquine administration. Although the proportion of recurrences estimated to be relapses following high-dose primaquine is low (~10-20% in these studies, depending on the transmission intensity) this suffices for the conditional probability of the relapse state to be high, when the recurrence occurs two months after treatment. This plot agrees with expert opinion and extensive historical data from artificial infection studies regarding how the probabilities of the three recurrence states would be expected to change as a function of the time-to-event.

Our goal was in part to capture, with a Weibull distribution, the time-to-event behaviour of early relapses due to temporal 'bunching'. We note that the time-to-event model does not claim to capture mechanistic properties of relapse but only empirical distributions of time to relapse. For this reason, extrapolation to different treatments with different post-prophylactic durations would require re-estimation of parameters.

- Data visualisation. The timing of recurrences are shown in Figure 4, but it's quite hard for the reader to mentally aggregate this data. Creating a histogram of this data, perhaps with classification would be quite helpful (possibly one for the Appendix).

We thank the reviewer for this excellent suggestion. We have added this to the supplementary information (Supplementary Figure 1).

- "The dynamics of time-to-relapse, irrespective of whether primaquine was administered, were explained as a 60/40 mixture between a highly periodic timing and a 'random' timing". Please provide a table (perhaps in the Supp Appendix) with the parameter estimates and uncertainty corresponding to this statement.

We have changed this sentence in the Results to include the uncertainty around the 60/40 split (first paragraph of section 2.2). In addition, the Supplementary Table 1 gives the posterior median and 95% credible intervals for the parameter q on which this statement is based.

- Sometimes it's unclear which estimate comes from which study: VHX, BPD, or joint.

We have gone through the paper and made sure every derived estimate is given the correct context, and have added "Unless explicitly stated, results are based on data from both trials combined." to the first paragraph of the Results section.

- Figure 4: Why is one of black triangles for relapses circled for less than 1 month. Was there any classification of recrudescence?

None of the recurrences were classified as recrudescences. This is consistent with other studies which indicate low grade chloroquine resistance (at most) and no piperazine or artemisinin resistance. Nevertheless, for consistency throughout the paper we now classify recurrences as reinfections or failures (relapses or recurrences). We have revised section 4.3.5 and Figure 4 to clarify this point. Construction of a model with a high sensitivity to recrudescence requires extra data on drug susceptibility. The timing of early relapse and recrudescence will be almost identical and the genetic

signatures can only falsify a recrudescence hypothesis rather than confirm it (both relapse and recrudescence can be clonal; relapse can also be sibling/unrelated, whereas recrudescence cannot).

The < 1 month circle was a mistake: delayed relapses were circled by matching to patient ID rather than episode ID, and so both infections in patient VHX 235 were accidentally circled. We thank the reviewer for spotting this mistake. We have also changed the point types in Figure 4 in order to avoid confusion between reinfection and failure.

- “Although recommended in most endemic countries, primaquine is not used widely because of the risks of iatrogenic haemolysis in patients with glucose-6-phosphate dehydrogenase (G6PD) deficiency”

This may be true in Asia, but not so in most of South America. In Brazil, for example, nearly 5 million people have been treated with primaquine without testing for G6PD deficiency since 2000.

We agree that South America is quite different in this aspect. However, the very large majority of vivax malaria cases (>90%) are in Asia, along with most of the more severe G6PD mutations (Mahidol, Viangchan, Canton, Vanua Lava etc.), as compared to the A- mutations prevalent in Africa and South America. Asia is where we believe the 8-aminoquinoline drugs have the greatest importance.

- The last paragraph of the Introduction is a bit out of place and reads as a summary of the results.

We have edited such that it now simply alerts the reader to the content of the paper, while also addressing some confusion around relatedness, in response to reviewer #2’s comments.

- The bottom panels of Figure 2 would be better on the same scale on the y-axis.

This has been changed. Thank you for suggesting this.

- There seem to be a mix of 80% and 95% credible intervals reported. Please stick with the more conventional 95% intervals.

We have amended this and the paper now only reports 95% credible intervals.

- The reinfection-adjusted estimates of primaquine efficacy are a particularly important contribution to the literature. Perhaps important enough to be noted in the abstract.

We have now included this in the abstract, but expressed as a failure rate (for consistency). This is stated as: “In this region of frequent relapsing *P. vivax*, failure rates after supervised high-dose primaquine are significantly lower (~3%) than estimated previously.”

- “If we assume the background transmission to have been constant over the 4 years studied (2010-2014)”. Constant at what rate? This information could be included in the suggested table describing the epidemiology of the trial sites.

We agree that this was not clear in the original submission. After discussion with the study PIs, it was decided that there could have been a significant decline in transmission intensity from the start of the VHX study up until the end of the BPD study. This possibility was corroborated subsequently by an

analysis of as yet unpublished data from a study that followed individuals who had a history of vivax and who were treated with high dose primaquine at the start of follow-up. These data showed an approximate halving of reinfection rates between 2010 and 2014 along the Thailand-Myanmar border region in proximity of Mae Sot. In addition, posterior predictive checks showed that the model was overestimating recurrence rates in the PMQ+ arm (comparisons between the observed time-to-recurrence data and data simulated under the posterior predictive distribution of model 2).

In light of this information, we decided to update the model parametrization so that differences in reinfection rates between the two studies could be formally specified. For simplicity, we did not model this as yearly reinfection rates but rather as study specific reinfection rates (the two studies were carried out sequentially). This estimated a decrease in the reinfection rate of approximately 50% between the two studies. This decrease matched the decrease estimated from our unpublished data (0.2 infections per 1000 years of follow-up in 2010 and 0.13 infections per 1000 years of follow-up in 2014). The posterior predictive checks (and the posterior predictive p-values) now do not show any large discrepancies between posterior predictive simulated data and the observed data (see Supplementary Figure 7).

- For the Bayesian methods described in Section 4.3.1, were there 6 or 8 chains?

Apologies for the confusion, there were 8 chains. The model script and the methods section were not consistent. We have rechecked all the statements for consistency between the RMarkdown file and the paper.

- Supplementary Figure 1: “The expected relatedness of meiotic siblings is the mean of a bimodal distribution with modes at 0.3 and 1 (see Appendix and 41,42 for explanation)”. Figure 2 of Wong et al gives modes of 0.33 and 1. Perhaps I’m missing something.

We thank the reviewer for spotting this mistake. It has been corrected so that it is now 0.33. We note that this is now Supplementary Fig 10.

Reviewer #2:

Leveraging two large vivax treatment trials carried out on the Thai-Myanmar border that involved >1000 patients, Aimee Taylor and colleagues have constructed two relapse prediction models based on 1) time-to-event data from ~1600 recurrences and 2) microsatellite genotyping data from ~500 paired recurrences in ~200 persons. They then combine these two models into a joint Bayesian model whereby the time-to-event probabilities serve as the prior estimates for the genetic model. This combined model finds that high-dose primaquine is highly effective in the region for radical cure/preventing relapse, bringing the re-infection adjusted failure rate down to 2-3% from a previous estimate of 10-15%. This research addresses a persistent problem in the vivax field – measuring the efficacy of primaquine in endemic settings – in a novel way. The findings, if true, are important and would be of widespread interest.

I cannot follow all the math, but my understanding is as follows. In Model 1, predictions are based on a higher rate of relapse after short-acting drugs (artesunate, chloroquine) and a higher likelihood of re-infection further out, with different rates/distributions for relapse vs. reinfection. In Model 2, between 3-9 microsatellite markers are used to categorize episodes genetically as clonal, sibling, or stranger to the previous episode based on pair-wise relatedness that is meant to reflect identity-by-descent (IBD) relatedness arising from meiosis of unrelated gametes in the mosquito. The model is driven by the key assumption that only relapses occur as siblings (but can be clonal or strangers), whereas re-infections occur only as strangers, and recrudescences are rare and clonal.

The overall scheme is clever in that it combines two disparate sources of predictive priors in a setting where many factors are relatively known – low transmission, lack of antimalarial resistance, intermediate complexity of infection, great genetic diversity despite low transmission, short relapsing strains, and common occurrence of heterologous relapses. The findings, that the vast majority of recurrences in SE Asia in the absence of primaquine treatment are due to relapse, is consistent with prior published estimates and expert opinion.

Thank you very much for this detailed summary of our work, elements of which we have used in our revision of section 4.3.3.

My primary question/concern is whether there is overinterpretation of limited genetic data leading to a false claim of accurate classification of IBD relationships. Another weakness that is less problematic is the uncertain applicability of the overall joint model to other areas where transmission is higher and antimalarial resistance rates are higher or unknown (this is acknowledged in the Discussion).

We partially agree that the joint model has uncertain applicability. The time-to-event model has a flexible parameterization (relapse is modelled as a mixture of an exponential and a Weibull distribution) whereby the prior specification is key to the identifiability of the posterior distribution. In a different setting, as long as there is some knowledge of the local epidemiology, this model will be easily applicable. The possibility of co-endemic highly periodic early and long latency relapse is not included in the model, but adding an extra component to the relapse mixture would be straightforward. As for the genetic model, its applicability is less certain. As we mention in the discussion, transmission intensity

and thus population genetic diversity are key factors: (i) our model cannot compute recurrence state estimates when complexity of infection is too high ($COI > 3$), and (ii) sufficient diversity is required in order to obtain meaningful results. The latter is an inherent limitation in genetic data of any modality.

However, the model does not over-interpret the available genetic data, especially regarding inference of relapse versus reinfection (further extensive explanation below). When the genetic data are few the model simply returns estimates close to the prior. For example, with a low number of markers in the simulated 'sibling' scenario (Figure 5), the model is undecided between reinfection and relapse; it only converges towards a confident estimation of the relapse state for more than 6 markers. Therefore, in the analysis of the VHX and BPD study data, for paired recurrences with few shared markers typed, this will result in a probabilistic estimate which is heavily informed by the time-to-event data and only slightly informed by the limited genetic data.

In the presence of error and minority clones, the model will have downwardly biased estimates of recrudescence (recrudescences will be estimated to be relapses). However, we do not believe this impacts our results on the Thailand-Myanmar border where there is at most only low grade resistance to chloroquine and no resistance to piperazine or artemisinin in *P. vivax*. Nevertheless, it is a limitation of the generalizability of the model and we clearly acknowledge this in the third paragraph of the Discussion with reference to our extensive simulation study described in more detail below.

In addition (and in response to reviewer #3), the penultimate paragraph of the discussion now explicitly lists the requirements for the application of this modelling approach.

Major comments

1. IBD classification based on pairwise genetic relatedness.

Introduction - Top of page 4 "Relatedness is based on identity-by-descent, an approach that accounts for parasites that are genetically identical by chance, ensuring a low false positive rate of relapse classification (2.2%)."

This sentence was confusing to me in the beginning – I could not understand what a low false positive rate of relapse classification meant. Even after reading everything, I find the sentence misleading because I would not think that pairwise relatedness of 3-9 microsatellite markers is enough to ensure accurate IBD classification.

We thank the reviewer for highlighting this point and apologize for the confusion caused. 1) To address the initial confusion and also the comment of reviewer #1 regards the misplaced nature of the paragraph in which this sentence featured, we have removed it. 2) We address the confusion about a) the meaning of low false positive rate and b) the utility of 3-9 markers for IBD classification in the following.

By low false positive rate we meant the proportion of inferred relapses in our null genetic data set. We have since modified the false positive rate to mean the proportion of inferred failures (relapse or recrudescence) in our null genetic data set. The null genetic data set is the set of all pairwise comparisons between episodes occurring in different people. Since relapses and recrudescences only occur intra-host, any failure inferred from our null genetic data set is a false positive. We explain this in the second paragraph of the main text, section 2.3, and for clarity we have changed 'false-positive rate' to 'false-failure discovery rate'. From the null genetic data we recover a false-failure discovery rate of

2.5%. This rate is low (lower than the customary false discovery rate associated with 95% confidence intervals). Geographical proximity may lead to infections in different individuals which are in fact highly related. Looking at how the diversity of these markers relate to geographical proximity is the subject of future work.

The genetic model does **not** estimate relatedness. Instead, within an intermediate step of the genetic model, it estimates the probability of the data given one of three proposed genetic relationships: stranger, sibling or clone. A formal description of this step can be found in the Supplementary text section 3.3.2. Informally, $\text{Prob}(\text{data given relationship}) = \text{Product of Prob}(\text{datum given relationship})$ where

$\text{Prob}(\text{datum given relationship}) = \text{Prob}(\text{datum given IBD}) \times \text{Prob}(\text{IBD given relationship}) +$

$\text{Prob}(\text{datum given not IBD}) \times \text{Prob}(\text{not IBD given relationship}).$

Where ‘relatedness’ is the probability of IBD between 0 and 1, while ‘relationship’ is either stranger, sibling or clone in this context. In terms of relatedness the equation above can be written,

$\text{Prob}(\text{datum given relationship}) = \text{Prob}(\text{datum given IBD}) \times \text{expected relatedness of relationship} +$
 $\text{Prob}(\text{datum given not IBD}) \times (1 - \text{expected relatedness of relationship}).$

We have made several revisions to clarify this point:

- Main text, section 4.3.3, includes the above description and first equation
- We have replaced ‘relatedness’ by ‘relationship’ where it is more accurate to do so (e.g. we now refer to graphs of relationships with edge relationship labels)

The utility of 3-9 markers to inform the genetic model with an intermediate step as described above was addressed by a simulation study, which we extended in response to the reviewers’ combined comments. The genetic simulations are described in detail below.

Genetic Simulations: to assess the data requirements for recurrent state inference, we simulated data under a relationship graph that contains a single between-episode edge representing a parasite haploid genotype with specified relationship of stranger, sibling or clone, which is among otherwise unrelated stranger parasites if the input COIs exceed one. This type of graph (in which noisy parasite haploid genotypes are unrelated) is the most diverse and thus most computationally challenging.

As described earlier for each of the stranger, sibling and clonal scenario, we simulated data for an initial and recurrent infection with respective COIs: 1 and 1, with and without error; 2 and 1, with and without error; 1 and 2, with and without error; 3 and 1, without error. As before, data were simulated for 3 to 12 markers. For the non-erroneous episodes with COIs of 1 and 2, we explored cardinalities of 4 (the minimum effective cardinality of any marker in the MS data that feature in the main text) and 13 (the mean effective cardinality of the MS data that feature in the main text). For the erroneous data and for the episodes with COIs of 3 and 1 we used cardinality = 13 only. The data and results can be found at the github repository under ‘Simulation_Study/SimulationStudy_Genetic_Model.Rmd’ and respective pdf. In the main text, details of the genetic simulation are provided in section 4.3.4.

Results: As noted earlier, when the effective cardinality is as high as 13 (the mean effective cardinality of the MS data that feature in the main text), nine markers provide sufficient information for reliable

recurrent state inference providing the COIs of the initial and recurrent infections are 1 and 1 respectively. In the main text, we briefly reference the following additional results. It is important to note that these results show the performance of the model when the prior puts equal weight on each of the three recurrence states. However, in the analysis of the VHX and BPD data, we make use of informative time-to-event data.

1. When data are few the genetic model distributes posterior probability density between potentially ambiguous states. Its confidence increases with the number of markers. This means that when data are few the genetic model is conservative and returns estimates close to the prior, guarding against over interpretation of limited data. When there are no data, the genetic model returns the prior (PriorReturnCheck_Genetic_Model.R). This is a useful internal quality control check for any Bayesian model.
2. Additional markers improve relapse and reinfection inference when the effective cardinality is low (Supplementary Fig 3a), when the data are highly erroneous (Supplementary Fig 3b), and when the COIs exceed one (Supplementary Fig 4); see revised section 2.5 main text. Specifically, three additional markers (totalling twelve) improve inference when the COI of the initial or recurrent infection increases to two (Supplementary Fig 4a). When the COI of initial infection increase to three, a small fraction of simulations under the sibling scenario have non-zero reinfection probability using twelve markers, suggesting a potential gain in precision with more markers (Fig S4b). To achieve zero reinfection probability under the sibling scenario when the data are very highly erroneous (e.g. 20% of markers are typed with error) more than 12 markers are required (Supplementary Fig 3b).
3. Under the model, clones are misinterpreted as siblings if the data are very highly erroneous, thereby hampering recrudescence inference. Previously, this was noted in the discussion and in section S3; Supplementary Fig 3b now illustrates this effect of genotyping error.
4. Supplementary Figure 4 now supports the pre-existing discussion around the frailty of recrudescence versus relapse inference in the presence of undetected minority clones (see caption).
5. Computational complexity biases the inference of recrudescence when infections highly diverse (Supplementary Fig 4b; see caption); this increases as more markers are typed. It is not a problem in the real field data. In the field data, complex infections most likely derive from either relapsing or co-inoculated parasites (there is little opportunity for superinfection due to active follow up in this low transmission setting). As such, they are liable to be interrelated and thus less diverse. It would, however, be a problem were the model used to analyse more markers. We thus refer briefly to this limitation in the discussion with reference to the Supplementary Figure 4.

If all genomic loci are considered, meiotic siblings would have pairwise relatedness = 0.5 (S2), but is the converse true when looking at a very small set of markers? – if 6 microsatellite markers covering only half the chromosomes (assume 3 of the 9 failed or were misclassified based on stutter peaks) had pairwise relatedness close to 0.5, does this mean the genotypes parasites are definitely siblings, and therefore relapse? In my understanding, the genetic model assumes that siblings are relapses rather than re-infection, but 0.5 pairwise relatedness at 6 repeat loci seems like it could represent re-infection

as well. How would the model interpret a recurrence with MOI of 3, containing a clone and two unrelated strains, but with overall pairwise relatedness close to 0.5?

Before answering the first question we paraphrase as follows: if a pairwise-comparison using 6 markers showed 3 identical and 3 different markers (identity-by-state = 0.5), does this imply the genotyped parasites are siblings and therefore relapsing? The answer to this question is no.

This is because we define relatedness as a pairwise probability of identity-by-descent (IBD), and not identity-by-state (IBS). IBD accounts for unrelated parasites that are IBS due to chance sharing of common allele, and thus is informed by the background population allele frequencies (see "Estimating relatedness between malaria parasites." *Genetics* (2019) for an in-depth comparison of the two approaches).

Because we define relatedness in terms of IBD, unrelated parasites that share by chance alleles for 3 out of 6 markers, i.e. IBS = 0.5 (or indeed all 6 of 6 markers, IBS = 1) have a non-zero probability of being unrelated and therefore a non-zero probability of being derived from a reinfection. For example, in the simulation studies the orange bars (probability of reinfection) in the "Stranger scenario" are high even when the fraction of markers at which evidence of IBS is close to 0.5 (e.g. in *SimulationStudy_Genetic_Model.pdf* compare left-top plot of Figure 1 and middle plot of Figure 5).

The important point around IBD versus IBS was not sufficiently clear previously. We apologise. To clarify, we state explicitly in the last paragraph of the introduction: "*We define relatedness as the pairwise probability of identity-by-descent (IBD), thus accounting for unrelated parasites that are identical-by-state (IBS) due to chance sharing of common alleles.*"

To alleviate the reviewer's concern that fewer markers will bias the results towards over-estimating the probability of relapse, we have plotted the relapse probability inferred in our null genetic data set as a function of the number of markers used (see Supplementary Figure 5). Again, it is important to note that these results show the performance of the model when the prior puts equal weight on each of the three recurrence states, whereas in the analysis of the VHX and BPD data we make use of informative time-to-event data. In Supplementary Figure 5, in the worst case scenario, where only one marker is used to estimate the recurrence state (1739 calculations in the null genetic data), the median posterior is equal to the prior value (set at 1/3 for each of the 3 recurrence states). In other words, the data are non-informative and the model returns the prior. When there are 2 or more shared markers, the median probability of the relapse state is always lower than the prior probability. For exactly 3 shared markers (177855 calculations) approximately 16% of the comparisons estimate a posterior probability of relapse greater than the prior value. In the case of 9 markers (55759 calculations) this is only 4%. Very few of these pairwise comparisons return relapse probabilities greater than our classification threshold: 2.6% and 1.4% for 3 and 9 markers, respectively.

2. 4.2 Microsatellite genotyping. Since the number of markers differed for different samples, please report somewhere how many pre- and post-treatment genotypes were found to be different vs. related.

In section 4.2, the classification 'related' versus 'different' was based on evidence of IBS and classification was applied to sample pairs, not genotypes. This was unclear. We apologize and have modified section 4.2 accordingly. Please see the caption of Supplementary Table 4 for pre- and post-

treatment classification counts expressed as the number of enrolment and recurrent episodes selected for additional marker typing at.

For the related pairs, how many were successfully genotyped at 6 additional microsatellite loci? (since the modeling suggests that 9 markers are needed to accurately classify siblings). How many of the “related” pairs were ultimately classified as clones vs. siblings vs. strangers – or what was the distribution of pairwise relatedness?

Regarding genotyping success, we have added counts to Supplementary Table 4. The genetic model outputs the probability that a specified recurrence is a recrudescence, relapse or reinfection (section 4.3.3). It does not output relationship classifications nor relatedness estimates. We thus report the numbers of samples genotyped at additional markers that were classified as relapse (also Supplementary Table 4). They might suggest that additional markers increase the likelihood of recurrent relapse. However, recurrences genotyped at additional markers were predominately obtained from patients in the VHX trial. They are thus more likely relapses due to treatment obtained. To properly understand the effect of additional markers on relapse inference, we partitioned the relapse probability inferred in the null genetic data by the number of markers used (see above).

If 10 samples were analysed in triplicate, how many were analyzed singly or in duplicate? If singly, then there is likely to be significant misclassification error due to slippage and stutter peaks.

Besides the 10 samples analysed in triplicate for quality checking, all samples were analysed singly. As such, there is likely to be some misclassification error due to slippage and stutter peaks. However, as we explain above and in reference to Supplementary Figure 3b, relapse versus reinfection inference is very robust to error. Recrudescence inference is not robust, but this is unlikely to impact our results given little evidence of *P. vivax* resistance on the Thailand-Myanmar border. In a setting where resistance is suspected, analysis in triplicate is merited.

3. Model details in Results. Since the novelty of the paper lies in the Bayesian models, the format of listing Methods in the last section doesn't serve the paper well, and I would've appreciated if more of the model design was incorporated into the Results section, rather than giving a brief review then reporting results which cannot be evaluated without diving into the supplementary material.

We thank the reviewer for recognising the methodological novelty of our work, but we strongly believe model details belong in the Methods section whose current placement is prescribed by the journal guidelines. Transfer of supplementary material into the Results section risks over-burdening a general reader. The current format provides an overview while also allowing an interested reader to delve deeper into the Methods and Supplemental Material when desired.

4. Time-to-event model. The assumptions in the genetic model and their implications are well laid out in the supplementary material. It would be nice to likewise list assumptions made in the time-to-event model.

We have added a section in the supplementary materials outlining the main assumptions of the time-to-event model, listed in order of approximate importance (see Supplementary Note, section 1).

Minor comments

5. 2.1 Overview of analysis – What were the “strongly informative priors for identifiability” of recrudescence and re-infections?

For recrudescence this was a nested mixing parameter of 1% with 95% credible interval 0.61%-1.64%. The rate parameter for time to recrudescence was a gamma with mean 0.01 (95 CI 0.08-0.12).

Re-infection was given a hierarchical mixing parameter with mean 30% (95% CI 14%-54%). The reinfection rate was given a gamma distribution with mean 0.0011 (95% CI 0.0008-0.0014). This corresponds to 1 reinfection every 900 days (95% CI 700-1200).

For extra clarity, we have added a supplementary Table (Supplementary Table 1) giving the exact specification of the priors and summary statistics of the posterior distributions.

6. Fig 5 – “The complexities of infection were 1 in both first and second infections.” I thought elsewhere, the models accounts for COI up to 3. Many studies have shown that vivax infections in SE Asia are more often polyclonal than not. Why is COI set =1 here?

The full suite of simulations showing the performance of the genetic model on synthetic data was difficult to present in a simple way and we decided to only show the results for COI=1. This highlights the overall trends for the number of markers required. However, for completeness we have now added results of simulations with higher enrolment COIs 2 and 3 (see Supplementary Figure 4 and `Simulation_Study/SimulationStudy_Genetic_Model.pdf`).

7. S7, p.31 – “4 ways two select two sporozoites” – typo

This has been corrected, thank you for pointing this out.

Reviewer #3:

For *P. falciparum* malaria, genotyping has been used to distinguish recrudescence and reinfection when assessing the efficacy of drugs aimed at blood-stage parasites. For *P. vivax*, dormant liver-stage parasites can lead to relapse. The possibility of relapse complicates the interpretation of drug trials since relapses may be clones or siblings of the blood-stage infection associated with the illness. This paper describes a method to combine information about the timing of recurrences and about genetic markers to provide estimates of the probability that a recurrence is a relapse, reinfection or recrudescence. This provides a more accurate picture of the efficacy of a liver-stage drug. The work is novel to my knowledge, and the paper is clearly and concisely written.

The estimates produced by this method are more sophisticated than just taking all recurrences together or by omitting only clear reinfections. It is plausible that the estimates are also more accurate, but this is not known and it may be that some assumptions are misleading (such as distributions being different to those assumed). There is some validation by simulation (section 2.5), but as far as I can work out, this assumes that all of the distributional assumptions are correct. How robust is the method if assumptions about, for example, seasonality are not correct? (Reinfection would usually be expected to be seasonal in a seasonal setting). Or if genotyping errors or imperfect detection of minority clones are incorporated?

We agree that the estimates of treatment efficacy are heavily model/assumption dependent. Given that we do not attempt to jointly estimate the probabilities of relapse, recrudescence and reinfection, it is possible to address concerns regarding model robustness with respect to misspecification by considering each model - the time-to-event model and the genetic model – separately.

Misspecification under the genetic model

An extended suite of simulations under the genetic model characterises the impact of misspecification in terms of imperfect detection of minority clones and genotyping errors.

Genetic Simulations: we simulated data under a relationship graph that contains a single between-episode edge representing a parasite haploid genotype with specified relationship of stranger, sibling or clone, which is among otherwise unrelated stranger parasites if the input COIs exceed one. This type of graph (in which noisy parasite haploid genotypes are unrelated) is the most highly diverse thus the most computationally challenging. For each of the stranger, sibling and clonal scenarios, we simulated data for an initial and recurrent infection with respective COIs: 1 and 1, with and without error; 2 and 1, with and without error; 1 and 2, with and without error; 3 and 1, without error. As before, data were simulated for 3 to 12 markers. For the non-erroneous episodes with COIs of 1 and 2, we explored cardinalities of 4 (the minimum effective cardinality of any marker in the MS data that feature in the main text) and 13 (the mean effective cardinality of the MS data that feature in the main text). For the erroneous data and for the episodes with COIs of 3 and 1 we used cardinality = 13 only. The data and full set of results can be found at the github repository under ``Simulation_Study/SimulationStudy_Genetic_Model.Rmd`` and respective pdf. In the revised main text, details of the genetic simulations can be found in section 4.3.4.

Results with a focus on model misspecification: as originally discussed, misspecification impacts severely recrudescence inference. This is because clones are misinterpreted as siblings in the presence of genotyping error (Supplementary Fig 3b). Supplementary Figure 4 now illustrates the pre-existing discussion around the frailty of recrudescence inference in the presence of experimentally undetected minority clones. This is outlined in more detail below, with reference to SimulationStudy_Genetic_Model.pdf.

Failure to experimentally detect data from a minority parasite haploid genotype will have different consequences depending on the relationship of the minority parasite haploid genotype in relation to other parasite haploid genotypes across episodes. For example, referring to the plots in Figures 6 and 7 of SimulationStudy_Genetic_Model.pdf as illustrative scenarios where 'COI x-y' denotes a COI of x in the first infection and a COI of y in the second infection,

- in the Sibling COI 2-1 case (left plot, Figure 7) failure to detect the stranger parasite will result in the Sibling COI 1-1 case (left plot, Figure 6), thereby increasing the probability of relapse; meanwhile, failure to detect the sibling parasite will result in the Stranger COI 1-1 case (middle plot, Figure 6), thereby decreasing the probability of relapse, but not erasing it. Note that the case in which the noisy parasite is unrelated demonstrates the most severe possible outcome: if the noisy parasite were related, failure to detect it would result in a Sibling COI 1 1 case thereby maintaining the probability of relapse.
- In the Stranger COI 2-1 case (middle plot, Figure 7) failure to detect either stranger parasite will result in the Stranger COI 1-1 case (middle plot, Figure 6), maintaining the probability of reinfection and relapse.
- In the Clone COI 2-1 case (right plot, Figure 7) failure to detect the stranger parasite will result in the Clone COI 1-1 case (right plot, Figure 7), thereby maintaining the probability of recrudescence and relapse; meanwhile, failure to detect the clonal parasite will result in the Stranger COI 1-1 case (middle plot, Figure 7), thereby replacing the probability of recrudescence with reinfection.

The examples above illustrate the robust versus frail nature of relapse inference versus recrudescence inference, respectively; we refer to them briefly in the caption of the Supplementary Fig 4. Relapse inference is also robust in the presence of error, whereas recrudescence is not.

In terms of genotyping error, Figures 9-11 in SimulationStudy_Genetic_Model.pdf show inference in the presence of unmodelled error. The probability of genotyping error, 0.2, was set extremely high to illustrate model behaviour clearly. Realistic error rates will have much less impact. Error largely impacts inference of recrudescence: in the 'Clonal scenario' clonal parasites are interpreted as sibling parasites and the probability of relapse tends towards one; Supplementary Fig 3b is mentioned in the main text as an example.

Misspecification under the time-to-event model

A major misspecification in the time-to-event model is the absence of seasonality for reinfection. Although we believe this to be important, a correct specification of seasonality would add considerable complexity to what are already complex models. We have, however, following this reviewer comment and a comment from reviewer 1, included a simulation study that examines the impact on parameter estimates of ignoring seasonality. The conclusion is that the impact is minimal with possible slight

underestimation of the reinfection rate and a slight overestimation of the proportion of relapses in the observed recurrences (Supplementary Fig 7). We emphasise that the time-to-event model is 'empirical': it does not attempt to model the time to relapse or reinfection mechanistically, but instead fits general distributional forms (Weibull and exponential) to observed times of recurrence, all the whilst formally considering censoring and previous treatments received.

In addition, the time-to-event model fully specifies a data generating process and therefore it is easy to generate synthetic data from the posterior predictive model. This can be used for two purposes: first, for qualitative checks of time-to-recurrence between the observed and simulated data; and second, to compute posterior predictive p-values on summary statistics of interest. The qualitative comparison shows a good fit between the posterior predictive of times to recurrence under the distributional assumptions of model 2 and the observed data (for an illustrative example see Figure 3 below). For the posterior predictive p-value calculation, we choose the recurrence rate per follow-up year as the summary statistic for each treatment arm. For the three arms the observed summary statistics are well inside the interquartile ranges of the posterior predictive summary statistics (Supplementary Fig 7). These encouraging results were obtained following some modifications of the time-to-event model (Model 2) after further investigation using simulation methods. Notably, in the updated model, we estimate directly the difference in transmission intensity between the two studies (transmission was approximately halved). This estimate is corroborated by unpublished data from the same clinics whereby individuals with a history of *P. vivax* infection were followed-up after high-dose primaquine administration.

In summary, we believe that the time-to-event model has sufficient complexity to capture accurately the dynamics of recrudescence, relapse and reinfection for short latency *P. vivax*. The posterior probabilities of the three possible 'states' derived from this model should be robust to misspecification. They most certainly provide more accurate estimates of treatment efficacy and reinfection rates than any other existing methods. In combination with genetic data (which unfortunately is not always informative), we would strongly argue that these results are robust enough to estimate reinfection-adjusted primaquine efficacy and key epidemiological parameters (e.g. reinfection rates).

Figure 3 Posterior predictive model checking. The synthetic data (red) were simulated under the parametric assumptions of model 2 using a random draw from the posterior distribution of the model fit to the pooled VHX and BPD data. We compare the empirical cumulative distributions of the synthetic and real data.

The numbers in Table 1 from the time-to-event analysis suggest that on average the rates of reinfection in the groups are estimated to be around 10 per 100py for ART and CQ and around 17 per 100py for primaquine+. A crude simple test assuming a Poisson distribution suggests that this is significantly higher. It is not obvious why the reinfection rate should be higher in the PQ arm - is it possible that this is an artefact of the time-to-event model rather than a chance finding?

A key strength of the time-to-event model is that it jointly analyses all the available timing data and borrows information across study arms. The reinfection rate in the model is a fixed-effect parameter (same for all individuals across the three treatment arms), now allowing for a decrease in reinfection rates between the VHX and BPD studies. However, a potential weakness in both the time-to-event and the genetic model is that neither allow for overlap of recurrence states: relapse, recrudescence and reinfection are considered mutually exclusive events. Individuals with large hypnozoite burdens who relapse frequently throughout the follow-up period could be reinfected at the same time as they relapse. Both the time-to-event and the genetic models would, on average, tend to label such an event as a relapse (depending of course on the timing and the genetic signature). Therefore, relapses can hide reinfection events (this is now mentioned in Supplementary Note 1). We describe in the following

paragraph how this assumption affects the time-to-event, and the interpretation of our Table 1 (in the revised version this is Table 2).

Under the assumption of mutually exclusive recurrence states, it follows that not all follow-up periods are equally informative. For example, more reinfections would be observed on average during 1 year of follow-up in an individual with a low relapse propensity (i.e. high reinfection propensity) as compared to an individual with high relapse propensity. Intuitively, this can be interpreted as individuals with high hypnozoite burdens will relapse before being reinfected, the relapse signal drowning the reinfection signal. In contrast, an individual with few or no hypnozoites approximately contributes the same follow-up information with respect to the reinfection rate as an individual who was given high-dose primaquine (i.e. had their liver cleared of hypnozoites). Whereas an individual with a large hypnozoite burden will comparatively not contribute much information regarding reinfection rates. The hierarchical/mixed-effects formulation allows for a direct estimation of these inter-individual differences in hypnozoite burdens and hence allows for a global estimate of the reinfection rate. The substantial heterogeneity in the propensity to relapse can be seen in the data: over 1 year of follow-up, some individuals who did not receive primaquine had no recurrent infections, while others had up to 14!

Participants have *P. vivax* illness at enrolment. It is not necessarily the case that illness only occurs following the primary infection, it may also occur as a consequence of relapse (from malaria therapy studies). The logic in the model seems to be that following enrolment subsequent relapses from the same inoculation follow a distribution, however the distribution may change depending on whether the illness visit was a consequence of a primary infection or a relapse.

No, this is not an assumption in the model. The time-to-event and the genetic model are both agnostic with respect to the true unknown recurrence state of the enrolment episode. However, there are two instances in which estimating the true enrolment state would be useful.

First, in our simulation of seasonal reinfection, we use the empirical distribution of enrolment times to estimate the seasonal fluctuations of the reinfection probability. We note that this is only to provide a reasonable model for seasonality as we do not report seasonal trends, and it allows a simulation-based analysis of the impact of model misspecification. However, it is certain that the time of year of enrolment provides some information regarding the recurrence state of the enrolment episode under a fully seasonal model.

Second, our model does not formally consider progressively lengthening times to relapse. Previous studies both in humans and monkeys that were not contaminated by possible reinfection (reviewed in White 2011) have all shown that the time intervals between consecutive relapses do tend to lengthen as the hypnozoite burden is depleted. In a model that takes seasonality into account it would be possible to estimate the probabilities of the recurrence state of the enrolment episode, and knowing the recurrence state of the enrolment episode would be informative as to the timings of the subsequent recurrences. However, this would require adding substantial extra complexity in the time-to-event model and we do not believe such granularity is necessary for robust estimation of the state probabilities of the observed recurrences.

A useful thing to know is what contribution the time information and the genetic information made in

the final overall estimate. Figure S2 shows this for individual points, but how does each component change the overall estimates of drug efficacy? There seems to be a big shift from the priors in Figure S4.

We agree that this is useful information. It is now included as a Supplementary Table 2.

As far as I can work out, the positions of the alleles on the chromosomes are not taken into account. For siblings, this could provide information in that breaks in chromosomes are more likely between alleles further apart than close together.

We agree that the positions of markers on chromosomes are important for dense marker sets (see Taylor et al. "Estimating relatedness between malaria parasites." *Genetics* (2019)). Methods that account for dependence between markers are predominately based on Hidden Markov Models (HMMs). Ignoring marker dependence can impair pairwise relatedness inference. However, the difference is negligible for sparse panels (24 or fewer markers). As such, accounting for marker positions in the present set of nine marker will almost certainly have a negligible effect on inference, and does not warrant the added complexity of the HMM framework. We have added a supplementary table detailing the positions of the markers (Supplementary Table 3).

Using the time-to-event data as a prior and the genetic data as the data should have a rationale. The justification on pS17 "it remains to be seen whether a formally joint model of both data types would add value" does not seem strong.

Jointly modelling two distinct data types formally is extremely difficult. There exists an entire field of research reported in the theoretical statistics literature which attempts to give decision-theoretic reasons for why or why not a joint model is optimal, see for example Jacob et al 2017 (reference 22 in the supplementary materials). No simple solution has been proposed thus far. Most applications that use disjoint modelling are justified informally, for example the entire PK-PD modelling field: PK models are first fit to observed drug concentrations and then the estimated drug time curves are used to predict PD outcomes (most often done with no propagation of uncertainty).

In this context, splitting the time-to-event data (which incorporates information regarding treatment) and genetic data (agnostic of treatment) is natural and should not induce any bias, but could theoretically reduce the information content. We propagate the uncertainty in the time-to-event model with 100 Monte Carlo runs. We do not see an easy and computationally tractable formulation that models both data types jointly. Therefore, this is a convenience approach with no obvious alternative.

There are some statements scattered in the discussion and supplementary information about what type of data would need to be collected and where the method would work. It would be helpful to have one paragraph with these collected together to make it easy to understand the circumstances in which this method would work. For example, the timing of monitoring to get accurate estimates of time-to-recurrence, or the background parasite diversity.

We have added a paragraph to the discussion (last paragraph)

Minor comments

Introduction. Last sentence of paragraph 1: "Reinfection rates will either be constant over time or seasonal". It is not obvious why they would be constant... there are very few sites with truly constant transmission.

This has been changed to: "Reinfection rates are usually seasonal".

The title talks of "resolving" the cause of recurrences but this claims too much, they cannot be resolved so much as have probabilities attached.

We agree with the reviewer that could sound like our work is overpromising. We have changed the title to:

Resolving the cause of recurrent Plasmodium vivax malaria probabilistically

'Resolving' was meant in the statistical sense whereby the data (genetic+time-to-event) are informative regarding the true unknown recurrence state. At the population level these probabilities will converge to the correct population estimates (e.g. giving correct drug efficacy estimates).

p8. Is it necessary to assume that the background transmission intensity was constant from 2010-2014? Could this not be estimated using either this data or another source?

This was also a concern raised by reviewer 1. We agree that this assumption is too restrictive. After discussion with the clinical study PIs, it was considered that there could have been a significant decline in transmission intensity between the start of the VHX study and the end of the BPD study. This possibility was corroborated subsequently by an analysis of as yet unpublished data from a study following individuals who had a history of vivax and who were treated with high dose primaquine at the start of follow-up. These data showed an approximate halving of reinfection rates between 2010 and 2014 along the Thailand-Myanmar border region in proximity of Mae Sot. In addition, posterior predictive checks showed that the model was overestimating recurrence rates in the PMQ+ arm (comparisons between the true recurrence time-to-event data and data simulated under the assumptions of model 2 whereby parameters were drawn from the posterior distribution of Model 2 fitted to the pooled data).

In light of this information, we decided to update the parametrisation of the model so that differences in reinfection rates between the two studies were formally considered. For simplicity, we did not model this as yearly reinfection rates but rather as study specific reinfection rates as the two studies were carried out sequentially. This estimated a decrease in reinfection rate of 50% between the two studies which matched the estimates from the unpublished data following high-dose primaquine treated individuals with a history of vivax. The posterior predictive checks (and the posterior predictive p-values) now do not show any large discrepancies between data simulated from model 2 under the posterior and the observed data.

p8. In section 2., it is stated that the reinfection-adjusted failure rate of primaquine is 2.6% (80% CI: 2.0-3.5), and this is compared to an estimate of 12% (80% CI 10-14). I could not work out if the 12% came

from one of the two studies or both (only one is referenced). This makes it trickier to compare the two estimates.

The 12% failure rate was estimated from the BPD study as it followed three times more patients treated with high-dose primaquine than the VHX study and almost every single recurrence was genotyped. For clarity we have changed this paragraph to include results from both the VHX and BPD primaquine treated patients. This now reads:

We estimated the reinfection-adjusted failure rate after supervised high-dose primaquine to be 3.0% (95% CI: 2.4-4.0) in the BPD study, 2.4% (1.7-3.3) in the VHX study, and 2.9% (2.3-3.8) in both studies combined. Of 853 patients who received supervised high-dose primaquine with 677 patient years follow-up (VHX and BPD combined), on average 2.5% (2.1-3.1) had at least one failure while they were followed up. This estimate (2.5%) is slightly lower than the overall failure-rate estimate (2.9%) because the overall failure rate accounts for loss to follow-up. In comparison, of 446 patients who did not receive high-dose primaquine with 330 patient years follow-up, 73.8% (62.4-76.6) had at least one failure during follow-up.

These estimates are based on the combined genetic and time-to-event model. For the BPD study, which contributed most of the data, the reinfection-adjusted estimate (3.0%) is significantly lower than the original reinfection unadjusted estimate of the failure rate at 12% (95% CI: 10-14)[19].

p13. The relapses are assumed to come from a mixture distribution with one part occurring 'at random'. The term 'random' is not specific. A random variable can follow a distribution, it just must have a stochastic element. It is the underlying distribution that would be important here.

We agree that the term 'random' can imply stochastic, especially in a general statistical context. However, here we were using it alongside reference to previous models, which have used a constant-rate model for the time-to-relapse, where the exponential distribution is the canonical model for stochastic continuous times to event. To clarify, have used quotation marks around 'at random' and have add (i.e. without periodicity) after its first mention.

p17. Section 4.3.3 I can see that the authors have tried to explain the genetic model without statistics. However, I think it could be made simpler and more intuitive.

Section 4.3.3 has been substantially re-worded. It is a little bit longer due to simpler but more intuitive and less technical phrases.

Could recrudescence plausibly happen at all 10 months after the clinical episode? (Figure 1). The probability is very low, but the decline might go to zero very quickly after a certain time interval.

We agree that recrudescence at 1 year should have a probability of zero. However, the choice of using an exponential distribution with a high rate parameter circumvents any need to choose a maximum time at which recrudescence could occur. As a consequence, every finite time-to-event will have a non-zero probability of being a recrudescence but the probabilities approach zero 'very fast', i.e. at an exponential rate. Non-zero probabilities of recrudescence at 10 months (the estimates are less than 10^{-10}) will have no impact on any of the results.

How drug studies use time data (eg a fixed time window) and genetic data (eg for *P. falciparum*) currently could helpfully be mentioned in a sentence in the introduction.

This is an important point. The current approach of using a fixed time interval and genetic analysis based on identity-by-state presents an opportunity for more resolved modelling akin to what we have done for *P. vivax* recurrence. We have added this point to the end of the penultimate paragraph of the discussion (as it presents a future opportunity and its comprehension relies on the prior introduction of concepts such as IBD versus IBS):

*The current framework (i.e. time-to-event plus genetic model that accounts for chance sharing of common alleles using an IBD-based approach) could be simplified and adapted for model-based distinction of recrudescence versus reinfection following treatment of *P. falciparum* in clinical trials that, at present, use a fixed time interval and IBS-based approach [7].*

REVIEWERS' COMMENTS:

Reviewer #1 (Remarks to the Author):

The authors have robustly responded to all of my concerns from my initial review. The new simulation studies are to be commended.

On behalf of my South American colleagues, I would like to push for the introduction to say that "primaquine is not widely used outside of South America".

Reviewer #2 (Remarks to the Author):

The authors more than adequately responded to all my comments. This is an impressive body of work.

Reviewer #3 (Remarks to the Author):

The authors have largely addressed my comments.

I have only a few comments.

1. In the last but one paragraph of the Discussion, the authors have listed the conditions where the method will work. Here, it could be mentioned that an idea of transmission over time is helpful (this changed the estimated reinfection rates) and that strongly informative priors are needed – and so information for the priors is likely to be useful. That relapses can hide reinfections may be relevant here too if they affect the estimates.

2. Assuming constant transmission when there is a seasonal pattern did not make much difference to the results, which is good. Did the authors try more than one seasonal pattern? It's possible that it did not matter much in the case of Thailand, but may potentially matter in a setting with a very strong pattern. If not, the wording should be more cautious.

Response to Reviewers

Reviewer 1

The authors have robustly responded to all of my concerns from my initial review. The new simulation studies are to be commended.

On behalf of my South American colleagues, I would like to push for the introduction to say that "primaquine is not widely used outside of South America".

We very much thank the reviewer for their kind words and for the time they invested in this review. We believe the reviewer's simulation requests considerably improved our work, thank you. We have added "*primaquine is not widely used outside of South America*" to the introduction as requested.

Reviewer 2

The authors more than adequately responded to all my comments. This is an impressive body of work.

We very much thank the reviewer for their kind words and their time invested in this review. Their suggestions greatly improved the quality of the work!

Reviewer 3

1. In the last but one paragraph of the Discussion, the authors have listed the conditions where the method will work. Here, it could be mentioned that an idea of transmission over time is helpful (this changed the estimated reinfection rates) and that strongly informative priors are needed – and so information for the priors is likely to be useful. That relapses can hide reinfections may be relevant here too if they affect the estimates.

First, we would like to thank the reviewer for their time invested in this manuscript. Their suggestions greatly improved the quality of the work!

Regards the comment above, we have changed the start of the penultimate Discussion paragraph to cover a broader overview of necessary prior information. This now includes the possibility of changing transmission over time along with the importance of strongly informative priors: "The time-to-event model also relies on understanding any temporal changes to transmission dynamics, as in the data analysed here; and on characterising the schizonticidal activity of the blood stage drugs (the terminal elimination half-life, which determines the continued suppression of parasite multiplication). This background knowledge must be encoded using strongly informative priors for identifiability of the parameters."

That relapses can hide reinfections is a limitation. We have thus revised the third paragraph of the discussion, which addresses with limitations: “A potential weakness in both the time-to-event and the genetic model is that neither allow for overlap of recurrence states. Individuals with large hypnozoite burdens who relapse frequently could be reinfected while they experience relapse. Both models would, on average, tend to label such an event as a relapse. Therefore, relapses can hide reinfection events.”.

2. Assuming constant transmission when there is a seasonal pattern did not make much difference to the results, which is good. Did the authors try more than one seasonal pattern? It's possible that it did not matter much in the case of Thailand, but may potentially matter in a setting with a very strong pattern. If not, the wording should be more cautious.

We only tried one temporal pattern. It is true that a more `spiked' temporal pattern could increase the bias noted in the simulation model. Since this is a limitation of the model, we have added the requested cautious statement to the third paragraph of the discussion: “we do not account for seasonality under the time-to-event model. Simulation suggests that this omission has little impact, but elsewhere it may have more bearing.”.